# Spectral analysis of climate dynamics with operator-theoretic approaches

Gary Froyland[1], Dimitrios Giannakis [2,3✉], Benjamin R. Lintner[4], Maxwell Pike[4] & Joanna Slawinska [5,6,7]

The Earth's climate system is a classical example of a multiscale, multiphysics dynamical system with an extremely large number of active degrees of freedom, exhibiting variability on scales ranging from micrometers and seconds in cloud microphysics, to thousands of kilometers and centuries in ocean dynamics. Yet, despite this dynamical complexity, climate dynamics is known to exhibit coherent modes of variability. A primary example is the El Niño Southern Oscillation (ENSO), the dominant mode of interannual (3–5 yr) variability in the climate system. The objective and robust characterization of this and other important phenomena presents a long-standing challenge in Earth system science, the resolution of which would lead to improved scientific understanding and prediction of climate dynamics, as well as assessment of their impacts on human and natural systems. Here, we show that the spectral theory of dynamical systems, combined with techniques from data science, provides an effective means for extracting coherent modes of climate variability from high-dimensional model and observational data, requiring no frequency prefiltering, but recovering multiple timescales and their interactions. Lifecycle composites of ENSO are shown to improve upon results from conventional indices in terms of dynamical consistency and physical interpretability. In addition, the role of combination modes between ENSO and the annual cycle in ENSO diversity is elucidated.

[1] School of Mathematics and Statistics, University of New South Wales, Sydney, NSW 2052, Australia. [2] Department of Mathematics and Center for Atmosphere Ocean Science, Courant Institute of Mathematical Sciences, New York University, New York, NY 10012, USA. [3] Department of Mathematics, Dartmouth College, Hanover, NH 03755, USA. [4] Department of Environmental Sciences, Rutgers, The State University of New Jersey, New Brunswick, NJ 08901, USA. [5] Center for Climate Physics, Institute for Basic Science (IBS), Busan, South Korea. [6] Pusan National University, Busan, South Korea. [7] Finnish Center for Artificial Intelligence, Department of Computer Science, University of Helsinki, 00560 Helsinki, Finland. ✉email: dimitrios.giannakis@dartmouth.edu

Ever since the discovery of phenomena such as ENSO[1] and the Madden-Julian Oscillation (MJO)[2], the objective identification and characterization of coherent modes of climate variability have been vigorously studied across the disciplines of Earth system science. In the face of dynamical complexity and event-to-event diversity, the state of large-scale patterns of climate dynamics is typically described through a reduced representation provided by climatic indices, constructed using physical understanding and/or statistical approaches. For example, ENSO is an oscillation with a broadband periodicity of 2–7 years, commonly monitored using so-called Niño indices[3]. The latter are defined as spatial and temporal averages of sea surface temperature (SST) anomalies over the equatorial Pacific source region of ENSO. Such indices are employed for a multitude of diagnostic and prognostic purposes, including lifecycle composites[4] and prediction skill assessment[5].

Clearly, the success of these efforts depends strongly on the properties of the indices employed to characterize the phenomenon of interest. In general, it is desirable that a climatic index be as objective as possible, i.e., reveal an intrinsic pattern of climate dynamics independent of subjective choices such as data prefiltering, or details of the observation modality. For oscillatory patterns such as ENSO and MJO, it is important that the indices reveal the full cycle as a sequence of observables, e.g., SST fields in the case of ENSO. Yet, despite their widespread use, conventional approaches for defining climatic indices have inherent limitations, obfuscating the properties of the phenomenon under study, and sometimes yielding inconsistent results[6]. Empirical Orthogonal Function (EOF) analysis[7], for example, is perhaps the most commonly used statistical technique for identification of climatic indices, yet it is well known to exhibit timescale mixing and poor physical interpretability due to EOF invariance under temporal permutations of the data, even in idealized settings[8]. In the context of ENSO, scalar Niño indices do not provide full information about the state of the cycle because the index could be increasing or decreasing.

In contrast to EOF analysis and related approaches, which identify patterns based on eigendecomposition of covariance operators, spectral analysis techniques for dynamical systems employ composition operators, such as Koopman and transfer operators[9–11]. A key advantage of this operator-theoretic formalism is that it transforms the nonlinear dynamics on phase space to linear dynamics on vector spaces of functions or distributions, enabling a wide variety of spectral techniques to be employed for coherent pattern extraction and forecasting. Indeed, starting from early spectral approximation techniques for Koopman[12,13] and transfer[14–16] operators in the 1990s, there has been vigorous research on operator-theoretic approaches applicable to broad classes of autonomous[17–22] and non-autonomous systems[23–28]. In addition, recently developed methods[29–37] combine Koopman and transfer operator theory with kernel methods for machine learning[38–40] to yield data-driven algorithms adept at approximating evolution operators and their spectra.

In this paper, we show that the operator-theoretic framework provides an effective route for identifying slowly decaying (equivalently, slowly decorrelating) observables of the climate system as dominant eigenfunctions of transfer/Koopman operators and their generator. These eigenfunctions directly describe coherent climate phenomena such as ENSO, with higher dynamical consistency and physical interpretability than indices derived through conventional approaches. The principal distinguishing aspects of this analysis, illustrated in Figs. 1, 2, 3, and 4, can be summarized as:

1. Identification of cycles from spatio-temporal information: Our spectral approach is based on dynamical systems techniques, providing a superior basis for extracting persistent cyclic behavior. We transform the underlying full nonlinear dynamics to a larger linear space, yielding a complete linear picture for our spectral analysis. This transformation is built directly from physical spatio-temporal fields such as SST snapshots. Complex pairs of eigenvalues and their eigenvectors directly reveal persistent cycles (see outer panels of Fig. 1) and their periods.

2. Dynamical rectification: The underlying oscillations in ENSO are clearly revealed in a "rectified" two-dimensional (2D) phase space provided by a complex eigenvector. Temporal evolution of the oscillation is well described by a harmonic oscillator, represented by motion at a fixed speed around a circle in 2D phase space, with the oscillation frequency $\alpha$ determined by the complex eigenvalue. Importantly, this property holds true even if the dynamics of the full system is chaotic. See Figs. 2, 3, and 4, and accompanying animations in Supplementary Movies 1 and 2 for illustrations. Our thorough treatment of rectification clearly shows the asymmetry of ENSO, and enables an estimate of the "local speed" of the ENSO cycle.

3. Phase equivariance: If the 2D phase space is partitioned into $S$ "wedges", each corresponding to a lifecycle phase, then the dynamical evolution of the samples starting in any given phase over a time interval of $2\pi/(S\alpha)$ maps them consistently to the next phase. See Fig. 1 (center) and Fig. 5 for examples of this behavior with $S = 8$. An important consequence of equivariance combined with slow decay is that it endows the identified phases with higher predictability, while enabling the discovery of new mechanistic relationships between physical fields because of a more accurate lifecycle. Our improved phasing suggests that ENSO has a more significant cyclical component than previously thought.

## Results

The perspective adopted here is to view a climatic time series $x_0, x_1, \ldots, x_{N-1} \in \mathbb{R}^d$ as an observable of an abstract dynamical system representing the evolution of the Earth's climate. That is, we envision that there is an (unobserved) state space $\Omega$ and a function $X : \Omega \to \mathbb{R}^d$ such that $x_n = X(\omega_n)$, where $\omega_n \in \Omega$ is the climate state underlying snapshot $x_n$. Moreover, we consider that there is an (unknown) dynamical evolution law $\Phi^t : \Omega \to \Omega$, such that $\Phi^t(\omega_0)$ is the climate state reached at time $t$ starting from an initial state $\omega_0$. In particular, the climate states underlying the observed data are given by $\omega_n = \Phi^{n\,\Delta t}(\omega_0)$, where $\Delta t$ is a fixed sampling interval. In the analyses that follow, $X$ will correspond to monthly averaged SST, sampled at $d$ Indo-Pacific gridpoints at a monthly sampling interval $\Delta t$.

Given the data $x_n$, our goal is to identify a collection of observables (eigenfunctions) $g_j : \Omega \to \mathbb{C}$ with two main features: cyclicity and slow correlation decay. First, the observables are cyclic in the sense that there is an associated period over which they approximately return to their original values. Second, the observables are slowly decaying (or "persistent" or "coherent") in the sense that their norm decreases slowly under forward evolution of the dynamics. In the context of this work, "slowly decaying" and "slowly decorrelating" observables are synonymous notions.

From a machine-learning perspective, this task corresponds to an unsupervised learning problem aiming to identify slowly decaying cyclic observables. Note that cyclicity is a significantly different objective than variance maximization performed in the Proper Orthogonal Decomposition (POD), EOF analysis, and related techniques[7,41,42]. Complex EOF analysis[43], Principal Oscillation Pattern (POP) analysis[44] and spectral analysis of autoregressive models[45] seek to identify oscillatory modes from time series, though generally through the restrictive lens of linear state space dynamics. Operator-theoretic approaches are able to

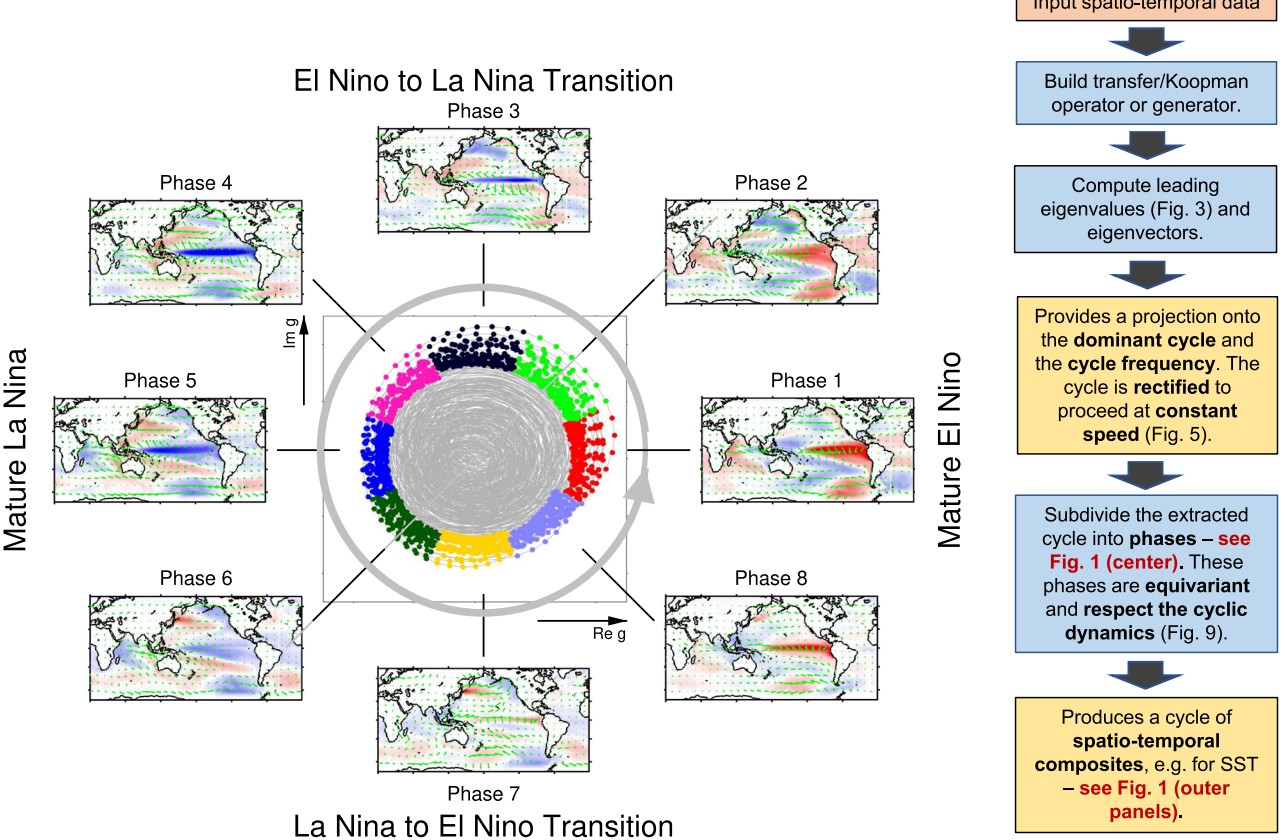

**Fig. 1 Operator-derived lifecycle of the El Niño Southern Oscillation (ENSO).** Left: Schematic representation of the canonical ENSO lifecycle recovered from a control integration of the Community Climate System Model version 4 (CCSM4). Center panel: 2D phase space associated with the real and imaginary parts of the eigenfunction $g$ of the generator representing ENSO. Each point in the 2D phase space represents an ENSO state. Dynamical evolution progresses in an approximately cyclical manner via counter-clockwise rotation. The period of the cycle is equal to $2\pi/\alpha \approx 4$ yr, where $\alpha$ is the imaginary part of the eigenvalue corresponding to $g$. The 2D phase space is partitioned into $S = 8$ "wedges" of equal angular extent (distinguished by different solid colors), each corresponding to a distinct ENSO phase. Outer panels: The panels linked to each wedge are phase composites of sea surface temperature (SST) (colors) and surface wind anomaly fields (green arrows). Collectively, they reveal a complete ENSO cycle, starting from a mature El Niño in Phase 1, and progressing to an El Niño to La Niña transition in Phase 3, mature La Niña in Phases 4–5, and La Niña to El Niño transition in Phase 7. The identified phases are equivariant, meaning that they each span a time interval of $2\pi/(S\alpha) \approx 0.5$ yr, and under forward dynamical evolution by 0.5 yr the samples making up phase $i$ correlate strongly with the samples making up phase $i + 1$. By virtue of this property, the generator-based ENSO lifecycle captures the duration asymmetry between the El Niño to La Niña and La Niña to El Niño transition. This is evidenced by the fact that the strongest La Niña anomalies occur in Phase 4, as opposed to Phase 5 (which would be expected for a time-symmetric oscillation). Right: Flow chart of the computational approach for identification of slowly decaying cycles through eigenfunctions of transfer/Koopman operators. Red-, blue-, and orange-shaded boxes represent input, computation, and output, respectively.

consistently extract cyclicity and coherence from nonlinear systems[30,46–48], without invoking a specific modeling ansatz such as linear dynamics.

In the present work, to cope with the high-dimensional data spaces resulting from climatic variables (e.g., SST fields), these operators will be learned using geometrical kernel methods combined with delay-embedding methodologies[49–51]. Delay-coordinate maps are also leveraged for analysis of climatic time series by extended EOF (EEOF) analysis[52], Singular Spectrum Analysis (SSA)[53–55], and related approaches, which extract temporal principal components (PCs) and associated spatiotemporal patterns (EEOFs) through singular value decomposition of a trajectory data matrix in delay-coordinate space. The success of these methods at recovering oscillatory patterns, including ENSO, has been interpreted from both state space[55] and operator-theoretic perspectives[33,37]. Ultimately, however, the extracted PCs from EEOF analysis/SSA are constrained to be linear functions of the delay-embedded data, and do not provide direct spectral information about evolution operators acting on observables.

**Operator-theoretic formalism**. Similar to classical methods such as EOF analysis, our approach assumes that the dynamics $\Phi^t$ on $\Omega$ is a stationary, ergodic process. We note that while the climate system is not strictly stationary, our methods perform well in extracting the dominant cycles on interannual or shorter time-scales. Mathematically, the stationarity is governed by a probability measure $\mu$ on $\Omega$, which is preserved by the dynamics; formally $\mu(\Phi^{-t}(A)) = \mu(A)$ for any measurable set $A \subseteq \Omega$. The ergodicity assumption is an indecomposability hypothesis: there are no non-trivial $\Phi^t$-invariant sets, meaning that $\Omega$ cannot be decomposed into separate subsystems.

Operator-theoretic approaches shift attention from studying the properties of the (generally, nonlinear) flow $\Phi^t$ on state space to studying its induced action on linear spaces of (generally, nonlinear) observables. We denote by $\mathscr{F}$ the space of complex-valued functions on $\Omega$. The space $\mathscr{F}$ has the structure of an infinite-dimensional linear (vector) space equipped with the standard operations of function addition and scalar multiplication, but the elements of $\mathscr{F}$ need not be linear functions. We will consider the

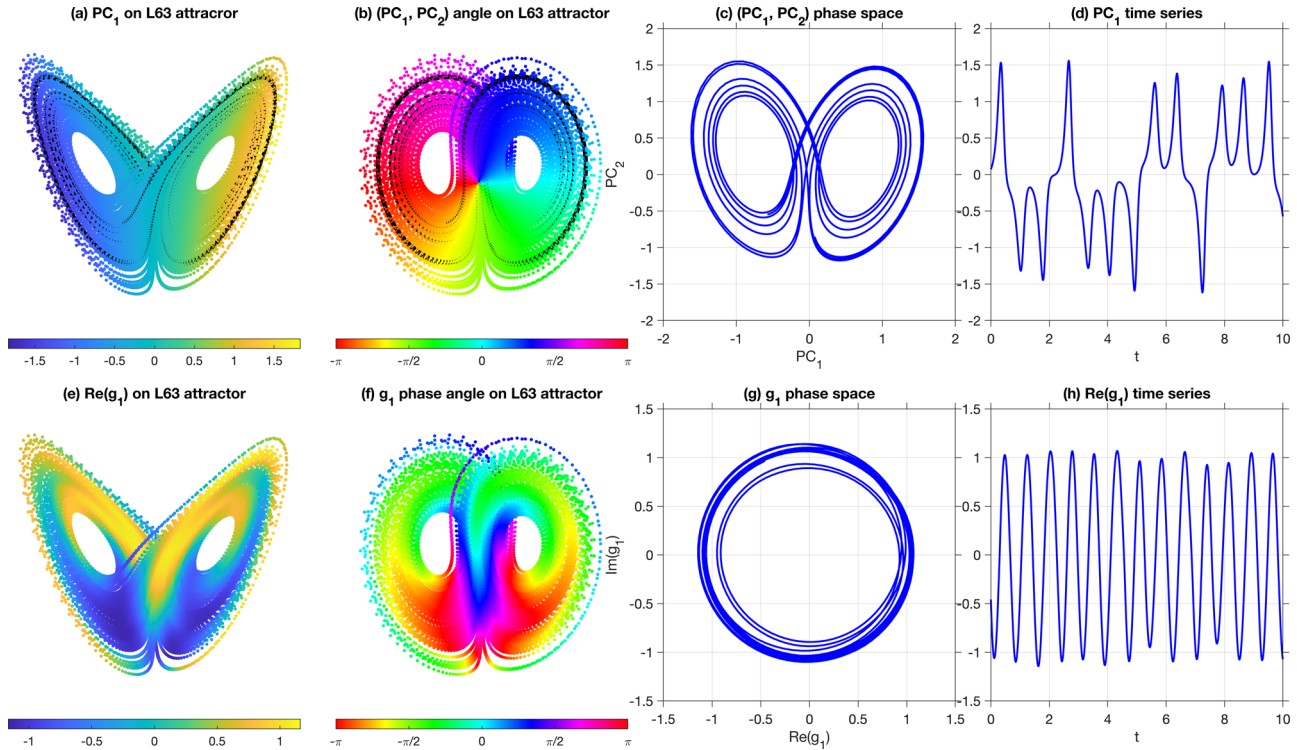

**Fig. 2 Comparison of EOF (covariance) eigenfunctions (a–d) and transfer operator eigenfunctions (e–h) for extraction of approximately cyclic observables of the Lorenz 63 (L63) chaotic system. a** The principal component (PC) corresponding to the leading EOF as a scatterplot (color is the EOF value) on the L63 attractor computed from a dataset of 16,000 points along a single L63 trajectory, sampled at an interval of $\Delta t = 0.01$ natural time units. The black line shows a portion of the dynamical trajectory spanning 10 time units, corresponding to the time series shown in **d**, **h** and phase portraits in **c**, **g**. **b** The phase angle on the attractor obtained by treating the leading two PCs as the real and imaginary parts of a complex observable. The black line depicts the same portion of the dynamical trajectory as the black line in **a**. **c** A 2D projection associated with the leading two EOF PCs for the same time interval as **d**. Since these PCs correspond to linear projections of the data onto the corresponding EOFs, the evolution in the 2D phase space spanned by $PC_1$, $PC_2$ has comparable complexity to the "raw" L63 dynamics, exhibiting a chaotic mixing of two cycles associated with the two lobes of the attractor. **e–h** The corresponding results to **a–d**, respectively, obtained from the leading non-constant eigenfunction $g_1$ of the transfer operator $P_\epsilon$ (see Methods). **f** The argument of the complex-valued $g_1$ (color is the argument) evaluated at the 16,000 points in the trajectory. Notice that there is a cyclic "rainbow" of color as one progresses around each individual L63 attractor wing in phase space. **g** Plots of these same arguments of $g_1$, now in the complex plane, demonstrating that the output of $g_1$ lies approximately on the unit circle. **h** The real part of the trajectory in **g** plotted versus time, illustrating approximately simple harmonic motion. Thus, the second eigenvector $g_1$ of the transfer operator $P_\epsilon$ extracts the dominant cyclic behavior of L63 on the attractor's wings.

subspace of observables $H = \{f \in \mathcal{F} : \int_\Omega |f|^2 \, d\mu < \infty\}$. Intuitively, thinking of $\mu$ as the climatological distribution of the system, the space $H$ consists of all observables with finite climatological mean and variance.

The dynamics acts naturally by composition on each element $f_0 \in H$. For invertible $\Phi^t$, the composition operator, $f_t := P^t f_0 = f_0 \circ \Phi^{-t}$, known as the transfer operator, evolves $f_0$ forward $t$ units of time to the function $f_t$. Dual to (and here, the inverse of) the transfer operator $P^t$, is the Koopman operator defined by $U^t f_0 := f_0 \circ \Phi^t$. Traveling forward in time along a trajectory $\{\Phi^t(\omega_0)\}_{t \geq 0}$, the observations recorded by $f_0$ along this trajectory are $f_0(\Phi^t(\omega_0)) = (U^t f_0)(\omega_0)$.

Ergodicity may be equivalently characterized by the constant function **1** being the unique (normalized) fixed point of $U^t$. Ergodicity implies (via Birkhoff's Ergodic Theorem or the strong law of large numbers) that sufficiently long trajectories in $\Omega$ will well sample $\mu$. This will be important in this paper because we are using a single trajectory as our input data. We note that many operator-theoretic algorithms may also use information from multiple trajectories and are not restricted to using a single time series. Similar operator constructions can be carried out in other functional settings, notably there is a well-developed spectral

theory for infinite compositions of different transfer operators arising from non-autonomous dynamical systems[24,56–58].

We now describe how the spectral properties of $P^t$ and $U^t$ provide natural notions of persistent almost-cyclic functions and observations. We distinguish between methods applicable for discrete- and continuous-time dynamics. Discrete-time approaches are based on approximations of the time-1 transfer/Koopman operators, whereas continuous-time approaches target the infinitesimal generators of the transfer/Koopman evolution semigroups. In the present setting of observables in the Hilbert space $H$ associated with the invariant measure, the Koopman and transfer operators are unitary, and are duals to one another under operator adjoints, i.e., $P^{t*} = U^t$. Thus, working with $P^t$ vs. $U^t$ is merely a matter of convention.

**Persistent cycles from the spectrum: discrete time**. Let $P = P^1$ be the time-1 transfer operator on $H$. If $Pg = \Lambda g$, with $g \not\equiv 0$, we call $\Lambda \in \mathbb{C}$ an eigenvalue and $g$ an eigenfunction. One has[11] that $|\Lambda| = 1$, $|g|$ is constant, and the collection of all eigenvalues of $P$, denoted $\sigma_e(P)$, is a subgroup of the unit circle (if $\Lambda, \hat{\Lambda} \in \sigma_e(P)$ then $\Lambda\hat{\Lambda}$ and $\Lambda/\hat{\Lambda}$ are both in $\sigma_e(P)$). As a simple example, if our phase space is $S^1$ (a circle of circumference $2\pi$) and $\Phi = \Phi^1$ rotates the circle by an angle $\alpha$, then $P$ has eigenvalues $\Lambda_k = e^{ik\alpha}$

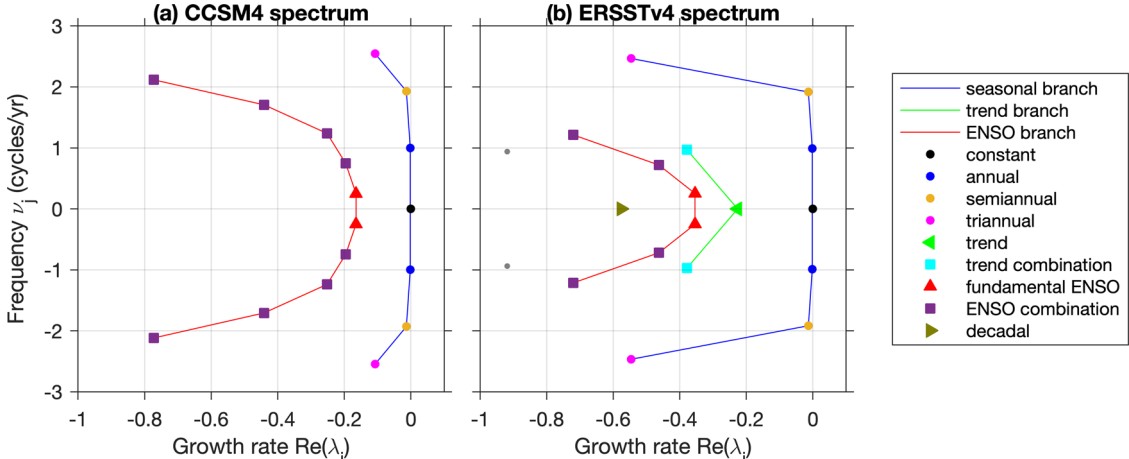

**Fig. 3 Leading generator eigenvalues $\lambda_j$ computed from CCSM4 (a) and ERSSTv4 (b) Indo-Pacific SST data, highlighting eigenvalues corresponding to seasonal, interannual, decadal, and trend modes.** The vertical and horizontal axes show the frequency $\nu_j = \mathrm{Im}\,\lambda_j/(2\pi)$ and growth rate, $\mathrm{Re}\,\lambda_j$, respectively. Note that complex eigenvalues occur in complex-conjugate pairs as appropriate for describing oscillatory signals at the corresponding eigenfrequencies. Moreover, negative values of $\mathrm{Re}\,\lambda_j$ correspond to decay. Lines connecting eigenvalues serve as visual guides for the seasonal (periodic), trend, and ENSO branches of the spectra. The annual (dark blue) and ENSO (red) eigenfrequencies indicated in the spectra correspond, through their imaginary parts, to the frequencies $\nu_{\mathrm{annual}}$ and $\nu_{\mathrm{ENSO}}$ discussed in the main text. Note that decadal modes are present in the CCSM4 spectrum, but they have larger decay rates, $-\mathrm{Re}\,\lambda_j$, than the range depicted in **a**. See Supplementary Table 3 for a listing of the leading 25 generator eigenvalues extracted from CCSM4 and ERSSTv4.

with corresponding eigenfunctions $e^{ik\theta}$ for $k \in \mathbb{Z}$ and $\theta \in S^1$. Analogous results hold for the Koopman operator $U = U^1$.

Numerical estimation of $P$ or $U$ inevitably introduces perturbations or "noise" to the operators, and leads to finite-dimensional representations which cannot exactly comply with the above theory. In particular, numerical representations of $P$ are often not unitary. Nevertheless, numerical schemes such as projected restrictions of $P$ or $U$ onto subspaces of $H$ spanned by locally supported or globally supported basis functions have been highly successful[17,23,29,59] and in certain settings, convergence results for the spectrum and eigenfunctions have been proven[14,15,22,60–62]. In these schemes, the spectrum of the approximate $P$ is contained in the unit disk $\{z \in \mathbb{C} : |z| \le 1\}$, rather than lying on the unit circle $\{z \in \mathbb{C} : |z| = 1\}$. This addition of noise, which may also be done theoretically, for example by convolution with a stochastic kernel[15,62,63], is frequently harnessed to easily select the most important eigenvalues from the typically infinite collection $\sigma_e(P)$, namely those eigenvalues with large magnitude (close to 1).

Let $P_\epsilon$ denote this perturbed operator and consider an eigenfunction $g^{(\epsilon)}$ corresponding to an eigenvalue $\Lambda^{(\epsilon)}$ of large magnitude. Because $(P_\epsilon)^t g^{(\epsilon)} = (\Lambda^{(\epsilon)})^t g^{(\epsilon)}$, these eigenfunctions $g^{(\epsilon)}$ decay slowly under iteration of $P_\epsilon$ relative to the decay rates of eigenfunctions corresponding to eigenvalues of smaller magnitude. It is these "leading" or "dominant" eigenfunctions that will persist over long timescales and will accurately describe the evolution of the dynamics over similarly long timescales.

Returning to our example $\Phi$ rotating the circle by an angle $\alpha$, the eigenfunctions of $P_\epsilon$ with least decay will be approximations of (and for carefully chosen approximations, equal to) $e^{\pm i\theta}$ ($k = \pm 1$) because they are the most regular, and persist longest under continued perturbation. The corresponding eigenvalues are $\Lambda^{(\epsilon)}_{\pm 1} = R_\epsilon e^{\pm i\alpha_\epsilon} \approx R_\epsilon e^{\pm i\alpha}$, for $0 < R_\epsilon \lesssim 1$, which correspond to rotation by $\pm\alpha$ with small decay rate of $R_\epsilon$ per unit time. Thus, the eigenvalues of $P_\epsilon$ of greatest magnitude (excluding the eigenvalue 1) automatically identify the rotation angle $\alpha$. See Methods for a description of our numerical approach for approximating $P$.

**Persistent cycles in continuous time**. In continuous time one can consider *generators* for the transfer and Koopman operators.

These generators are time-derivatives of $P^t$ and $U^t$, and are given by $Gf = \lim_{t\to 0} \frac{1}{t}(P^t f - f)$ and $Vf = \lim_{t\to 0} \frac{1}{t}(U^t f - f)$, respectively. The operators $G$ and $V$ are defined on a dense subspace of $H$, and are skew-symmetric duals to one another, i.e., $G = V^* = -V$. One has that $\sigma_e(G) = \sigma_e(V)$ are additive subgroups of $i\mathbb{R}$ (the eigenspectrum lies on the imaginary axis in $\mathbb{C}$); that is, if $\lambda, \hat{\lambda} \in \sigma_e(G) = \sigma_e(V)$ then $\lambda + \hat{\lambda}$ and $\lambda - \hat{\lambda}$ are both in $\sigma_e(P)$. Eigenvalues of $G$ and $V$ are interpreted as rates of rotation per unit time. If our phase space is $S^1$, and $\Phi^t$ rotates the circle at a rate $\alpha$, then $G$ and $V$ have eigenvalues $\lambda_k = \pm ik\alpha$ and corresponding eigenfunctions $e^{ik\theta}$ for $k \in \mathbb{Z}$ and $\theta \in S^1$.

The operators $G$ and $V$ "generate" the semigroup of operators $P^t$ and $U^t$ by $P^t = e^{tG}$ and $U^t = e^{tV}$, and the spectral mapping theorem connects their spectra: $\sigma_e(P^t) = e^{t\sigma_e(G)}$ and $\sigma_e(U^t) = e^{t\sigma_e(V)}$ (if $\Phi^t$ is not invertible, the spectral value $0 = e^{-\infty}$ is treated separately). For example, the relationship $\Lambda = e^\lambda$ links the eigenvalues $\Lambda$ of the discrete-time operators with the eigenvalues $\lambda$ of their continuous-time counterparts.

As in the discrete-time setting, one may perturb the generators by addition of a diffusion process or through a numerical scheme. In the former case, if $\Phi^t$ is governed by a vector field then natural "diffused" versions $G_\epsilon$ and $V_\epsilon$ of $G$ and $V$ are provided by normalized forward and backward Kolmogorov equations, respectively. In the latter case, one may apply various numerical schemes[26,30,34,36,64]. The scheme[34] is outlined in the Methods section. The eigenvalues of $G_\epsilon$ and $V_\epsilon$ are in general complex numbers with zero or negative real part. For the same reasons as in the discrete-time setting, one seeks eigenvalues with real part closest to the imaginary axis, which describe the slowest decay rate. In our example of a circle rotation with rotation rate $\alpha$, the eigenfunctions of least decay rate are $e^{\pm i\theta}$ ($k = \pm 1$) with corresponding eigenvalues $\lambda^{(\epsilon)}_{\pm 1} = -r_\epsilon \pm i\alpha_\epsilon \approx -r_\epsilon \pm i\alpha$, for $r_\epsilon \lesssim 0$ ($r_\epsilon$ is analogous to $\log R_\epsilon$ from the discrete-time setting).

**Eigenvalue frequency analysis of monthly-averaged Indo-Pacific SST**. We analyze model and observational SST data over the Indo-Pacific domain 28°E–70°W, 60°S–20°N. This domain was selected as a representative region of activity for several large-scale modes of climate variability on seasonal to decadal

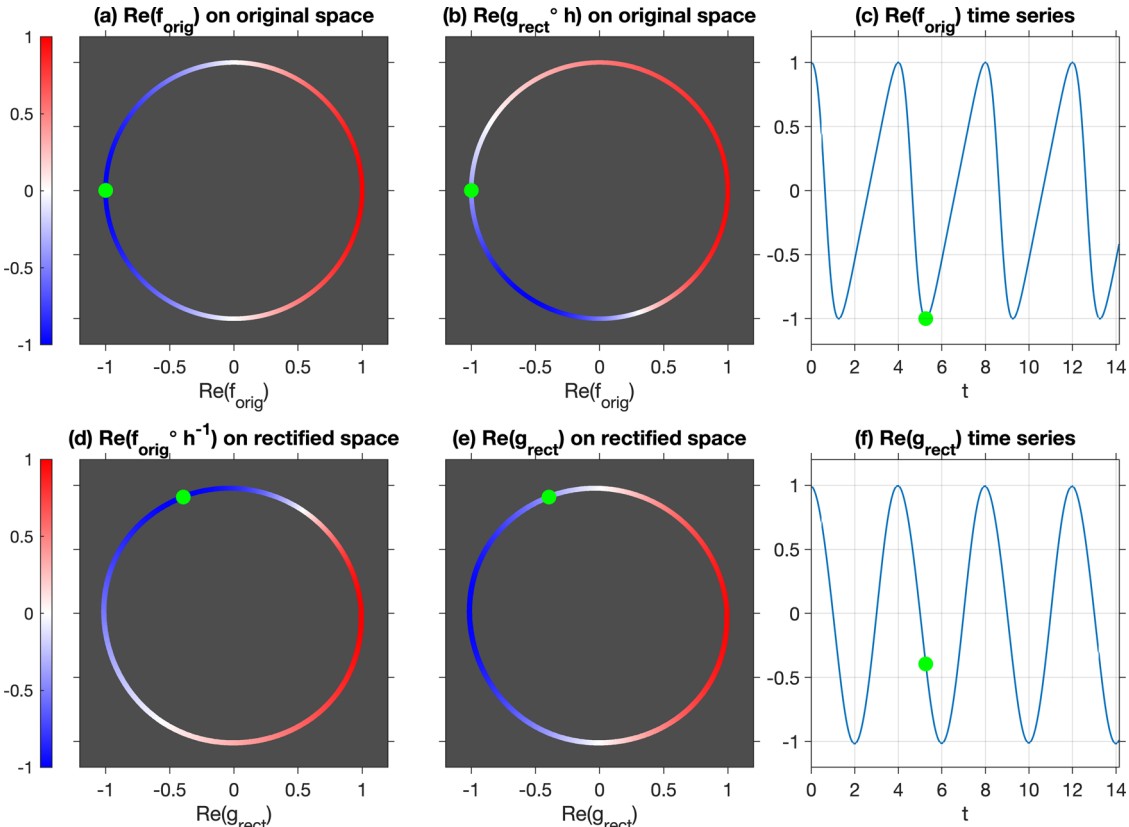

**Fig. 4 Rectification of a variable-speed oscillator by eigenfunctions of the generator.** The dynamics is chosen such that the speed $\frac{d\theta}{dt}$ is faster when $\theta$ lies in the interval $\Theta_{\text{fast}} := (0, \pi)$ and slower for $\theta \in \Theta_{\text{slow}} := (\pi, 2\pi)$, resulting in the time series for $\cos(\theta(t))$ shown in **c**. **a** The state space of the original oscillator, i.e., the unit circle $S^1$ consisting of all phase angles $\theta \in [0, 2\pi)$, colored by the value of the real part of the function $f_{\text{orig}}(\theta) = e^{i\theta}$, namely $\text{Re}\, f_{\text{orig}}(\theta) = \cos\theta$. In an analogy with ENSO, $\theta = 0$ and $\pi$ (green dot) would correspond to El Niño and La Niña climate states, respectively, while $\cos\theta$ would correspond to a Niño index. The asymmetry in rotation speed then mimics the fact that El Niño–La Niña transitions take a shorter amount of time than La Niña–El Niño transitions. In **d** we push down to rectified state space and show in color the real part of $f_{\text{orig}} \circ h^{-1}$. Note that La Niña (green dot) appears earlier in this constant-speed cycle on rectified space; this compensates for the variations of speed of the original oscillator. It is this representation, i.e., the original "ENSO index" mapped to the rectified state space, that we focus on in Fig. 5. **e** The real part of the eigenfunction $g_{\text{rect}}(\theta') = e^{i\theta'}$ (color) on rectified space, which appears as a pure cosine wave in **f**. The evolution of the phase angle in the rectified state space is that of a harmonic oscillator with constant angular frequency $2\pi/T$; note the period is $T = 4$ years in analogy with ENSO. Finally, in **b** we pull the function $g_{\text{rect}}$ back to the original space and display the real part of $g_{\text{rect}} \circ h$ (color).

timescales, including ENSO, ENSO combination modes[65], and the Interdecadal Pacific Oscillation (IPO)[66]. The model data comprise 1300 yr of monthly-averaged SST fields from a pre-industrial control integration of the Community Climate System Model Version 4 (CCSM4)[67], sampled at the model's native ocean grid of approximately 1° resolution. As observational data, we use monthly averaged SST fields at 2° resolution from the Extended Reconstructed Sea Surface Temperature Version 4 (ERSSTv4) reanalysis product[68] over the period January 1970 to February 2020. The resulting SST data vectors $x_n$ have dimension $d = 44{,}771$ and 4868 for CCSM4 and ERSSTv4, respectively.

Our numerical approach builds an approximation of the generator in a data-driven basis consisting of eigenvectors of a kernel matrix. The kernel matrix, $\boldsymbol{K}$, has size $\tilde{N} \times \tilde{N}$, where $\tilde{N} = N - Q + 1$, and is constructed from delay-embedded SST fields over a window of $Q - 1 = 48$ lags of length $\Delta t = 1$ month, corresponding to an interannual time interval of $Q\,\Delta t = 4$ years. Its eigenvectors represent temporal patterns that can be thought of as nonlinear generalizations of the PCs obtained via EEOF analysis. Previously[69–71], such kernel eigenvectors were shown to successfully recover physically meaningful modes from monthly averaged SST data in both Indo-Pacific and Antarctic domains. For our purposes, however, the eigenvectors and eigenvalues of $\boldsymbol{K}$

are employed to construct a data-driven version of the regularized generator $V_\epsilon$, and extract dominant modes by solution of an associated eigenvalue problem (see Methods). The approximation basis formed by the eigenvectors of $\boldsymbol{K}$ is (i) learned from the high-dimensional SST data at a feasible computational cost; (ii) is refinable, in the sense of having a well-defined asymptotic limit as the amount of data $N$ increases; and (iii) as the delay window $Q\,\Delta t$ increases, it is provably well-adapted to representing eigenfunctions of the generator[33,37]. The results we obtain are not particularly sensitive to the precise choice of kernels and lags, nor to the use of the generator or transfer operator. For example, similar results can be obtained with the transfer operator $P^{\Delta t}$ constructed using a single lag ($Q = 2$) of length $\ell = 12$ months (see Methods). A summary of the dataset attributes and numerical parameters employed in our computations is displayed in Supplementary Tables 1 and 2.

Figure 6 and Supplementary Table 3 show eigenvalues $\lambda_0, \lambda_1, \ldots$ of the generator $V_\epsilon$ computed from the CCSM4 and ERSSTv4 datasets, arranged in order of decreasing real part (i.e., increasing decay rate). The leading eigenvalues form distinct branches corresponding to (i) the annual cycle and its harmonics; (ii) ENSO and its combination modes with the annual cycle; and (iii) low-frequency (decadal) modes with vanishing oscillatory

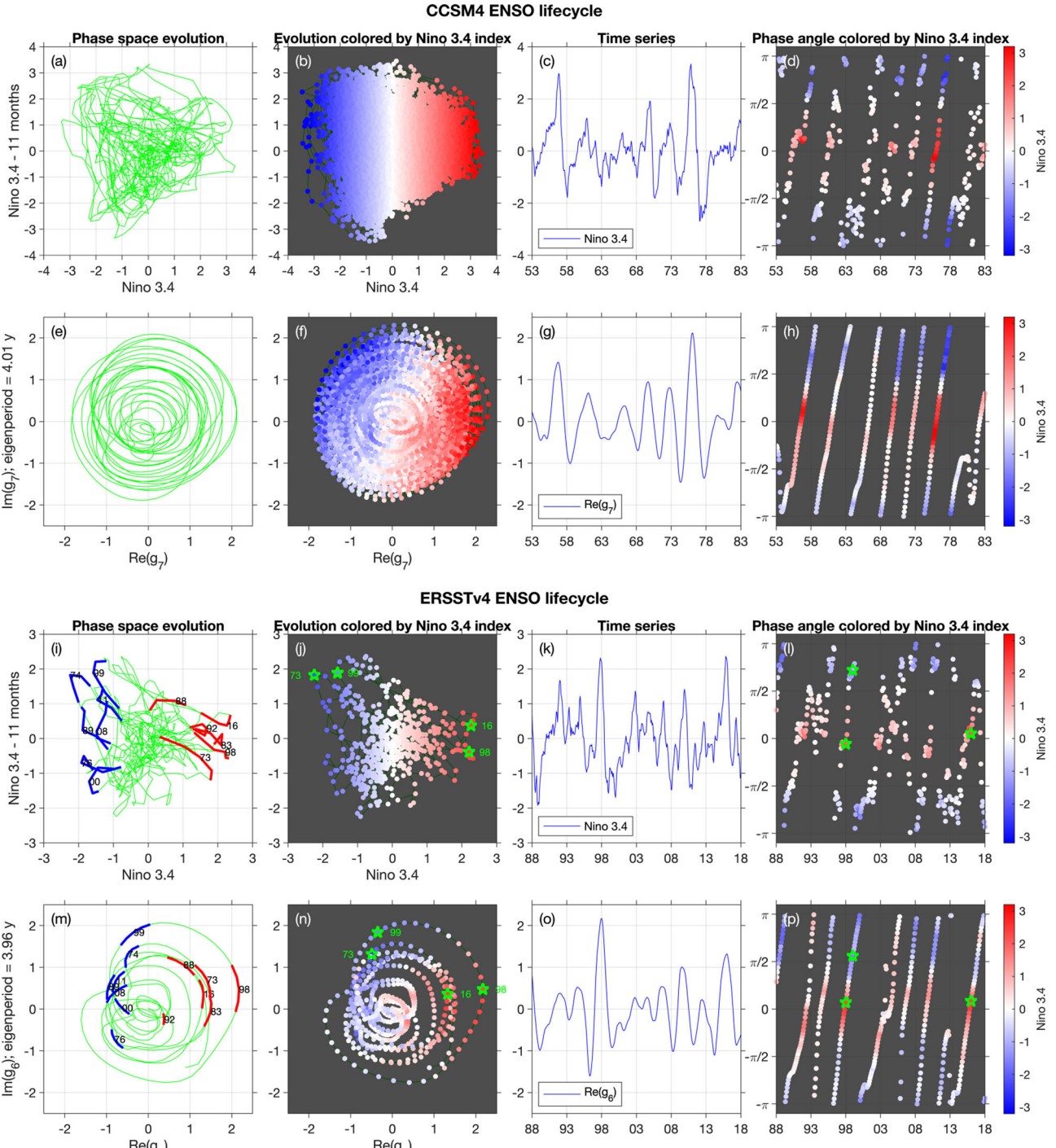

**Fig. 5 ENSO lifecycle for CCSM4 (a–h) and ERSSTv4 (i–p), reconstructed using lagged Niño 3.4 indices $f_{nino}$ and the complex eigenfunction $g_j$ of the generator. a**, **e**, **i**, **m** The evolution of the ENSO state in the 2D phase spaces determined from the Niño- (**a**, **i**) and generator-based (**e**, **m**) indices. For clarity of visualization, in **a** we show the evolution over a 100-year portion of the 1300-year dataset. Significant historical El Niño and La Niña events are marked in red and blue lines in **m** for reference. **b**, **j** (resp. (**f**, **n**)) show scatterplots of the original (resp. rectified) Niño- and generator-based lifecycle colored by the Niño 3.4 index. These plots are analogous to the original and rectified oscillator plots in Fig. 4 (**d**, **e**), respectively. **c**, **g**, **k**, **o** The Niño 3.4 time series (**c**, **k**) and the real part of $g_j$ (**g**, **o**), plotted over a 30-year portion of the available data. These time series are analogous to those in Fig. 4 (**c**, **f**) respectively. **d**, **h**, **l**, **p** The phase angles determined from $f_{nino}$ (**d**, **l**) and $g_j$ (**h**, **p**). Note that the slope in **h**, **p** is approximately constant, consistent with the automatic rectification process, namely that trajectories precess around the origin at a fixed angular speed. This regular rectification from our complex eigenvector is in strong contrast to the irregular angular behavior of the lagged Niño 3.4 index in **d**, **l**.

frequency. In the case of the observational data, the spectrum also contains a trend-like mode representing climate change, as well as combination modes representing the modulation of the annual cycle by the trend (see Supplementary Fig. 1).

In interpreting the results in Fig. 6 and Supplementary Table 3, it should be kept in mind that, modulo a small amount of numerical drift, the CCSM4 data are generated by autonomous dynamics associated with fixed (pre-industrial) concentrations of

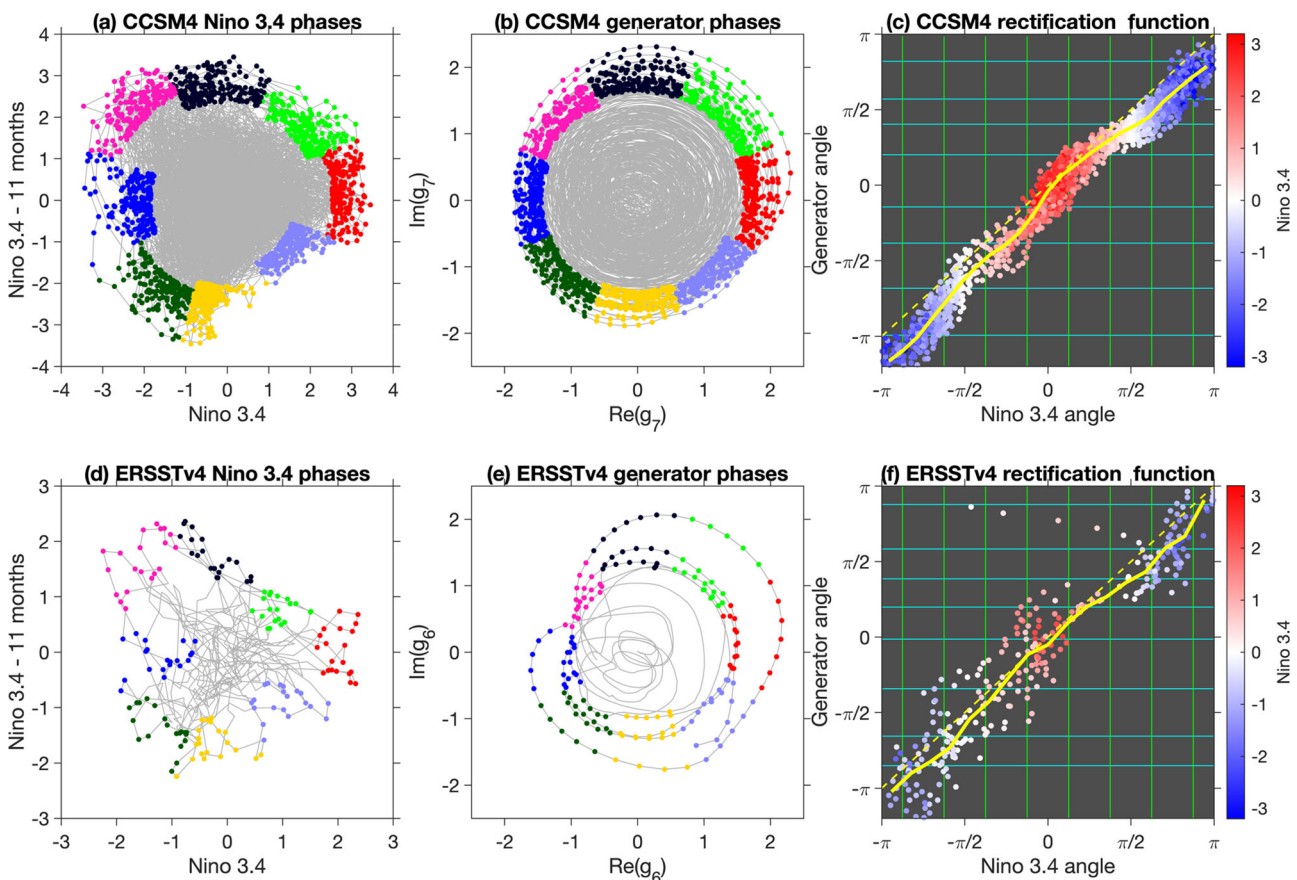

**Fig. 6 ENSO phases for CCSM4 (a, b) and ERSSTv4 (d, e), identified using lagged Niño 3.4 indices $f_{nino}$ (a, d) and generator eigenfunctions $g_j$ (b, e).** Phases are selected by partitioning the 2D phase space into eight angular sectors of uniform angular extent (equal to 45°), and then selecting the samples with the $m$ largest distances (corresponding to ENSO amplitudes) from the origin in each sector. We use $m = 200$ and $m = 20$ for CCSM4 and ERSSTv4, which corresponds to ~1.5% and 3% of the available data per phase, respectively. The selected data points in each phase are marked by distinct colors, with red corresponding to the El Niño phase, Phase 1. Progression from Phase 1 to Phase 8 takes place in a counter-clockwise sense. The La Niña phases in the Niño 3.4 and generator representation are Phases 5 (blue) and 4 (pink), respectively. Gray lines show the phase space evolution over the entire 1300-year (**a**, **b**) and 50-year (**d**, **e**) analysis intervals. **c** and **f** The rectified generator angle (y-axis) against the Niño-3.4 angle (x-axis). The yellow curves fit the data points and by construction pass through the origin (0,0), which corresponds to El Niño for both the Niño-3.4 and generator representations. Note that according to the yellow curve in **c**, El Niño for the Niño-3.4 representation, occurring at angle $\pi$ on the x-axis, corresponds to an angle of $\approx 3\pi/4$ on the y-axis for the rectified cycle. This quantifies the more rapid transition between El Niño and La Niña in CCSM4, in comparison to the reverse transition. Although less noticeable in the yellow curve in **f**, the ERSSTv4 results display a similar El NiñoLa Niña transition asymmetry, manifested by the tendency of La Niñas (dark blue dots) to occur below the diagonal dashed line. The speed of transition is indicated at the finer level of phases by the green lines, which are spaced equally on the x-axis according to Niño-3.4 phase boundaries. Wider (resp. narrower) spacing of the horizontal green lines corresponds to a slower (resp. faster) transition between phases.

greenhouse gases and perfectly periodic radiative forcing representing the seasonal cycle. In particular, the phase of the seasonal cycle is implicitly represented in the delay-embedded SST data. The autonomous techniques employed in this paper are therefore rigorously applicable in this dataset; see Supplementary Note 1 for further details. In contrast, the ERSSTv4 data are subject to different natural and anthropogenic external forcings (e.g., volcanoes and greenhouse gas emissions, respectively), so strictly speaking our autonomous methodology does not formally apply here. Nevertheless, our spectral decomposition separates the trend (corresponding to a real eigenvalue) from the periodic and approximately periodic cycles (corresponding to complex eigenvalues), which are by definition trendless. In fact, we posit that an advantage of our approach is that it is capable of extracting trendless cyclical modes in ERSSTv4 without ad hoc detrending of the data, which is oftentimes performed in the context of EOF analysis and related approaches.

In both CCSM4 and ERSSTv4, the seasonal-cycle modes occur first in our ordering, which is consistent with the fact that these are purely periodic modes remaining correlated for arbitrarily long times. Two pairs of eigenfrequencies $\nu_j := \mathrm{Im}\,\lambda_j/(2\pi)$ in this family are accurately identified by the data-driven eigenvalue problem, namely the annual (1 yr$^{-1}$) and semiannual (2 yr$^{-1}$), eigenfrequencies where the numerical results agree with the true values to within 1% and 4%, respectively (see Supplementary Table 3). The third (triannual) harmonic is not identified as accurately, being assigned an eigenfrequency of $\simeq 2.5$ yr$^{-1}$ as opposed to the expected 3 yr$^{-1}$. This discrepancy is at least partly due to finite-difference errors in our numerical approximation of the generator; this is discussed in more detail in the Methods section. Other contributing factors to approximation errors for the eigenfrequencies include the Nyquist limit (which imposes a limit of $1/(2\,\Delta t) = 6$ cycles/yr on the maximum frequency that can be resolved with a monthly sampling interval) and the addition of diffusion (which in general perturbs

the eigenvalues along both the real and imaginary axes) in the construction of the regularized generator $V_\varepsilon$.

Beyond the seasonal cycle branch, the CCSM4 spectrum exhibits a branch of eigenvalues consisting of a pair of fundamental modes with an interannual frequency $\nu_7 \simeq 0.25$ yr$^{-1} =: \nu_{\text{ENSO}}$, as well as combination frequencies $\nu_j$, $j = 9, 11, 13, 15$, approximately equal to $\nu_{\text{ENSO}} + m\nu_{\text{annual}}$, where $\nu_{\text{annual}} = 1$ yr$^{-1}$ is the annual-cycle frequency, and $m$ is an integer taking values in the set $\{-2, -1, 1, 2\}$. Note that the spacing of 2 in the index $j$ is due to restricting to positive frequencies in this discussion; see Supplementary Table 3.

We will shortly interpret the eigenfunction corresponding to the eigenvalue $\nu_{\text{ENSO}}$ as representing the fundamental ENSO cycle. We note that this choice is unambiguous as the eigenvalue with largest real part and frequency close to 0.25 yr$^{-1}$. Similarly, the frequencies $\nu_{\text{ENSO}} + m\nu_{\text{annual}}$ are naturally interpretable as combination modes, consistent with the group structure of the generator spectrum described above. Further notable aspects of these results are that (i) distinct generator eigenvalues correspond to distinct combination frequencies (as opposed to EOF analysis, which mixes the combination and fundamental frequencies[65]); and (ii) two harmonics are identified corresponding to the annual and semiannual cycles. We have verified that the ENSO eigenfrequencies extracted from the CCSM4 data remain unchanged to two significant digits for embedding windows ranging from 1 year ($Q = 12$) to 16 years ($Q = 192$); see Supplementary Table 4.

ENSO and ENSO combination eigenvalues are also identified in the ERSSTv4 spectrum, but these eigenvalues occur after an eigenvalue with vanishing imaginary part that we interpret as a representation of climate change trend. As shown in Supplementary Fig. 1(a), the eigenfunction time series corresponding to this eigenvalue has a manifestly nonstationary character, which is broadly consistent with accepted climate change signals such as persistent warming from the 1980s to early 2000s, "hiatus" during the mid to late 2000s, and accelerated warming during the early to mid 2010s[72]. In addition, the trend eigenfunction time series is found to correlate with area-averaged anomalies of Indo-Pacific SST and global surface air temperature, with 0.83 and 0.78 correlation coefficients, respectively. As with ENSO, this trend eigenfunction comes with its own "combination frequencies" close to 1 yr$^{-1}$ (since the trend frequency is zero), capturing the modulation of the annual cycle by the trend (see Supplementary Fig. 1(b)). Aside from this trend family, both the CCSM4 and ERSSTv4 spectra contain additional modes with zero corresponding eigenfrequency, representing internal decadal variability of the Indo-Pacific[69,70]. The spectra also contain interannual modes with higher frequencies than $\nu_{\text{ENSO}}$, notably a mode with an approximately 3-year eigenperiod (see Supplementary Table 3). In what follows, we will focus on the fundamental ENSO eigenfunctions and the corresponding lifecycle analysis. These correspond to eigenfunctions $g_7$ and $g_6$ in the CCSM4 and ERSSTv4 ordering, respectively. Since the observational data are sparser and noisier than the model data, we expect larger (numerical) decay rates for the observational data as stronger diffusion is needed to regularize the generator (see Methods). This is borne out in Fig. 6, where the real parts of the generator eigenvalues for ENSO and the ENSO combination modes are more negative for the ERSSTv4 data than for CCSM4.

In summary, we have extracted ENSO eigenfunctions and eigenfrequencies from two datasets (CCSM4 and ERSSTv4), using two computational techniques (generator and transfer operator) and a range of numerical parameters (lag and embedding window length). Moreover, in each spectral analysis experiment, there is no ambiguity in associating particular eigenfunctions with ENSO, as discussed above.

**Rectified cycles from eigenfunctions.** As described above, when nonzero eigenfrequencies exist, the dominant eigenfunctions correspond to observables with approximately cyclic evolution, even if the underlying flow $\Phi^t$ is aperiodic. Below, we will use this idea to extract a rectified ENSO lifecycle from the spatiotemporal SST data. We first describe the mathematical construction, using idealized dynamical systems as examples.

Let $(U_\epsilon)^t g^{(\epsilon)} = (\Lambda^{(\epsilon)})^t g^{(\epsilon)}$ as before. We follow an orbit in state space $\Omega$ starting at some $\omega_0$. Evaluating both sides of $(U_\epsilon)^t g^{(\epsilon)} = (\Lambda^{(\epsilon)})^t g^{(\epsilon)}$ at $\omega_0$, we obtain $((U_\epsilon)^t g^{(\epsilon)})(\omega_0) \approx g^{(\epsilon)}(\Phi^t(\omega_0)) = (\Lambda^{(\epsilon)})^t g^{(\epsilon)}(\omega_0)$, where we have inserted the definition of $U_0^t$ as the middle term, recalling we have $U_\epsilon \approx U$ in some sense. Defining the multiplicative action of a complex number $\Lambda$ on another complex number by $M_\Lambda : \mathbb{C} \to \mathbb{C}$ by $M_\Lambda z = \Lambda z$, we have that $g^{(\epsilon)}(\Phi^t(\omega_0)) \approx M_{\Lambda^{(\epsilon)}}^t (g^{(\epsilon)}(\omega_0))$. Thus, we may think of the eigenfunction $g^{(\epsilon)}$ as an approximate projection (or factor map) from $\Omega$ to $\mathbb{C}$; this is summarized in the following (approximate) commutative diagram:

$$
\begin{array}{ccc}
\Omega & \xrightarrow{\Phi^t} & \Omega \\
{\scriptstyle g^{(\varepsilon)}}\downarrow & & \downarrow{\scriptstyle g^{(\varepsilon)}} \\
\mathbb{C} & \xrightarrow[M_{\Lambda^{(\varepsilon)}}^t]{} & \mathbb{C}
\end{array}
$$

Evolution under $\Phi^t$ on $\Omega$ is projected down (by $g^{(\epsilon)}$) to approximately a fixed multiplicative action on $\mathbb{C}$ by $\Lambda^{(\epsilon)}$. Further, for $|\Lambda^{(\epsilon)}| \approx 1$, we may consider the multiplicative action of $M_{\Lambda^{(\epsilon)}}$ as an approximate action on $S^1 := \{z \in \mathbb{C} : |z| = 1\}$. Recalling that $\Lambda_{\pm 1}^{(\epsilon)} \approx R_\epsilon e^{\pm i\alpha}$ with $0 < R_\epsilon \lessapprox 1$, the multiplicative action of $M_{\Lambda_{\pm 1}^{(\epsilon)}}$ corresponds to an approximate rotation on $S^1$ by an angle of $\pm\alpha$. Thus, for $|\Lambda_{\pm 1}^{(\epsilon)}| \approx 1$, evolution under $\Phi^t$ on $\Omega$ is projected down (by $g^{(\epsilon)}$) to approximately a fixed rotation on $S^1$ by $\alpha$. The above statement is illustrated numerically for the Lorenz equations in Fig. 2(e–g), where $g^{(\epsilon)}(\Phi^t(\omega_0))$ is plotted for $t \in [0, 160]$. The evolution lies approximately on $S^1 \subset \mathbb{C}$ (Fig. 2(g)), rotating at an approximately fixed rate (Fig. 2(h)). See Supplementary Movie 1 for a more direct visualization of these results. Projections of this type for real eigenfunctions of the transfer operator have been used to project out fast dynamics in multiple time scale systems[73].

The fact that the rotation on $S^1$ occurs at close to a fixed rate is a key aspect of our ENSO analysis, and so we emphasize this property by discussing a simple example that is strongly illustrative for climate cycles such as ENSO. We imagine a crude model of the ENSO cycle with one-dimensional phase space $\Omega = S^1$. The dynamics of this idealized model is given by a flow $\Phi^t : S^1 \to S^1$, generated by a nonconstant velocity on $S^1$. We choose a sawtooth-like velocity field to model the observation that the La Niña to El Niño transition is slower than the transition in the other direction[74]; see Fig. 4(c) for the corresponding evolution of a "normalized Niño 3.4" index vs. time and Supplementary Movie 2 for the corresponding animation. In this situation there is no need to "extract" a cycle in the dynamics, because the dynamics is a cycle—but importantly with nonconstant speed.

Similar to above, let $M_\Lambda^t$ denote the flow that advances the angle on $S^1$ by $\arg(\Lambda)t$, where $\arg(\Lambda) = (2\pi)/T$ and $T$ is the period of the cycle. The flow $M_\Lambda^t$ has a constant velocity around $S^1$, namely $2\pi/T$; see Fig. 4(f) for the corresponding cosine-like evolution. Because $\Phi^t$ and $M_\Lambda^t$ are both cycles of the same period, there exists a homeomorphism $h : S^1 \to S^1$ conjugating $\Phi^t$ and $M_\Lambda^t$; that is, $h \circ \Phi^t = M_\Lambda^t \circ h$, summarized in the commutative

diagram below:

$$S^1 \xrightarrow{\Phi^t} S^1$$
$$h \downarrow \qquad\qquad \downarrow h$$
$$S^1 \xrightarrow{M_\Lambda^t} S^1$$

We denote by $\theta$ the angle in the "original" cycle (the upper part of the commutative diagram) and by $\theta'$ the angle in the rectified cycle (the lower part of the commutative diagram); by definition $\theta' = h(\theta)$. We set $\theta = 0$ to represent the peak El Niño state in our crude cyclic model of ENSO, and without loss of generality we fix $h(0) = 0$ so that peak El Niño occurs at the same angle $\theta = \theta' = 0$ in both the original and rectified cycles. We define $\theta = \pi$ as peak La Niña, according to the original cycle, directly opposing El Niño; this is represented by the green dot in Fig. 4(a).

The eigenfunction of the Koopman operator corresponding to $M_\Lambda^t$ with eigenvalue $\Lambda$ is $g_{\rm rect}(\theta') := e^{i\theta'}$; this eigenfunction is illustrated Fig. 4(e) where the value of the real part of $e^{i\theta'}$ is colored. By the above conjugacy, the function $(g_{\rm rect} \circ h)(\theta) = e^{ih(\theta)}$ is an eigenfunction of $U^t$ with eigenvalue $\Lambda$; see Fig. 4(a). Because La Niña is reached more quickly from El Niño than vice-versa in the original flow, so too (by conjugacy) must this occur in the rectified, constant-speed flow. Thus, La Niña in fact appears earlier than half-way through the rectified cycle; see the green dot in Fig. 4(e), which lies at the angle $h(\pi)$. Finally, let $f_{\rm orig}(\theta) := e^{i\theta}$ represent the complex-valued function corresponding to our crude cyclic model of ENSO, where $\theta = 0$ is El Niño and $\theta = \pi$ is La Niña. We can map $f_{\rm orig}$ to the rectified space by $f_{\rm orig} \circ h^{-1}(\theta') = e^{ih^{-1}(\theta')}$; the real part of this latter function is shown in Fig. 4(b). We will use versions of the functions $f_{\rm orig}$ and $f_{\rm orig} \circ h^{-1}$ as our main demonstration of our rectification process in Fig. 5. These results are an example of the automatic rectification performed by Koopman eigenfunctions (Theorem 17.11[11]) for systems with discrete spectra. Operator-theoretic approaches to different kinds of rectification have also been explored[75,76].

**A rectified ENSO lifecycle from eigenfunctions**. We now apply the ideas from the previous two subsections to the CCSM4 and ERSSTv4 data, where $g_{\rm rect}$ in those sections will be the generator eigenfunctions $g_7$ and $g_6$, arising from these two datasets, respectively. In the following we will refer to $g_6$ and $g_7$ collectively as simply $g_j$. Figure 5 compares several aspects of the $g_j$ to new, lagged ENSO indices $f_{\rm nino}$ derived from the Niño 3.4 index output from the CCSM4 and ERSSTv4 data as follows. At each time instance, $f_{\rm nino}$ is a 2D vector consisting of the current Niño 3.4 value and its value $\ell$ months in the past; that is, (Niño 3.4$(t)$, Niño 3.4$(t - \ell$ months)). We choose $\ell$ to be the lag that gives the most cycle-like behavior for $f_{\rm nino}$. If the Niño 3.4 index evolved as a perfect cycle with a period of $T = 4\ell$ months, the two components of $f_{\rm nino}$ would be in quadrature (90° phase difference), resulting in a purely angular motion in the associated 2D phase space. This situation would be analogous to the evolution of the $f_{\rm orig}$ observable depicted in Fig. 4(a), which is periodic but not of fixed frequency. Yet, in Fig. 5(a, i), it is evident that the evolution of $f_{\rm nino}$ exhibits significant departures from an ~4-year cycle, featuring both retrograde and radial motion, particularly in the case of the ERSSTv4 data (Fig. 5i). In Fig. 5(d, l), we show the evolution of the phase angle obtained by treating the components of $f_{\rm nino}$ as the real and imaginary parts of a complex number, analogous to the L63 example in Fig. 2(b, f) (note that the latter representations are in the full phase space). Here, an approximately cyclical evolution of $f_{\rm nino}$ would induce an approximately

monotonic phase evolution (modulo $2\pi$), which would additionally be linear for a constant-frequency cycle. While such a behavior is discernible in Fig. 5(d, i), the phase evolution of $f_{\rm nino}$ is clearly corrupted by high-frequency noise due to retrograde/radial motion.

Consider now the generator eigenfunctions $g_j$. The time series plots (Fig. 5(c, g, k, o)) demonstrate that the real part of $g_j$ is positively correlated with the Niño 3.4 index (the first component of $f_{\rm nino}$): large positive values of Re $g_j$ tend to coincide with large positive values of the first component of $f_{\rm nino}$, including a number of significant events in the recent observational record such as the 1997/98 and 2015/16 El Niños. Recall that despite the presence of a climate-change signal in the ERSSTv4 data, the extracted ENSO eigenfunctions are trendless. Figure 5(f, n) displays scatterplots of the 2D phase spaces associated with the real and imaginary parts of $g_j$, colored by the Niño 3.4 index. These plots are analogous to the scatterplots of Re $(f_{\rm orig} \circ h^{-1})$ in Fig. 4(d), and illustrate that the very negative Niño 3.4 index values (deep blue) occur not directly opposite the very positive Niño 3.4 index values (deep red), but instead appear earlier in the rectified cycle. These facts and the fact that the corresponding eigenfrequencies $v_j$ are interannual and well-approximate $v_{\rm ENSO}$, provide evidence that the $g_j$ provide a representation of the ENSO lifecycle; a fact which will be corroborated further below using phase composites. Before doing that, however, we note two important aspects of the results in Fig. 5.

First, the generator eigenfunctions provide a significantly more cyclic representation of the ENSO lifecycle than conventional Niño indices. In Fig. 5e, m, the 2D phase space trajectories associated with the real and imaginary parts of $g_j$ are seen to undergo a predominantly polar evolution, with little to no retrograde motion when $g_j$ is located sufficiently away from the origin ($|g_j| \gtrsim 1$). As noted above, this is in contrast to the retrograde and radial motion seen in the Niño 3.4-based $f_{\rm nino}$ index. Moreover, in separate calculations we have verified that the generator eigenfunctions $g_j$ are also more cyclical than the two-dimensional $f_{\rm nino}$ indices constructed from the Niño 4, 3, and 1+2 indices. Two-dimensional phase space representations of the ENSO state with approximately cyclical behavior can also be constructed through multivariate indices, such as SST and thermocline depth anomalies[4], that reveal recharge–discharge processes[77], but these representations are also generally less coherent than those provided by the generator eigenfunctions.

Second, the generator eigenfunctions "rectify" the ENSO cycle in a manner analogous to the oscillator example in Fig. 4. In Fig. 5(h), the phase angle associated with the CCSM4-derived $g_j$ undergoes a near-linear evolution, with some excursions from this behavior occurring. We observe that these deviations from linear behavior occur when the Niño 3.4 (scalar) index is close to zero (white color in Fig. 5(h, p)). Mathematically, deviations from cyclic behavior are more likely when $|g_j|$ is small, which implies Re $g_j$ is also small, and is in turn consistent with weak ENSO amplitude. Visually, the rectification induced by $g_j$ can be seen in the time series plots in Fig. 5(c, g), where a comparatively uniform El Niño–La Niña cycling of Re $g_j$ (Fig. 5(g)) is contrasted with slow La Niña to El Niño ramp ups followed by rapid El Niño to La Niña decays in the $f_{\rm nino}$ representation (Fig. 5(c)). In Fig. 6(c), we examine the relationship between the phase angles associated with $f_{\rm nino}$ and $g_j$ through a curve fit of $\theta' := \arg g_j$ as a function of $\theta := \arg f_{\rm nino}$ (shown in a solid yellow line). The fitted curve provides an estimate of the homeomorphism function $h$ discussed above in the context of the oscillator example. When $\theta = \pi$ (i.e., during La Niñas according to the Niño 3.4 index), the fitted $\theta'$ is less than $\pi$, which shows that La Niña events occur earlier than half-way through the $g_j$ cycle, as in Fig. 4.

A similar general behavior of the phase angle is observed for the ERSSTv4 data (Figs. 5(l, p) and 6(f)), though as one might expect the results are noisier than for CCSM4. Still, the phase angle progression associated with $g_j$ (Fig. 5(p)) exhibits a significantly more rectified behavior than its $f_{nino}$ counterpart (Fig. 5(l)), particularly during significant El Niño/La Niña events (highlighted with green star markers). Interestingly, the generator angle $\arg g_j$ corresponding to La Niña events following strong El Niños (e.g., the 1973/74 and 1999/00 La Niñas in Fig. 5(n)) is close to 90°. This is consistent with the fact that strong consecutive El Niño and La Niña events in the observational record have a tendency to occur one year apart, corresponding to a quarter of the 4-year ENSO cycle.

In summary, our spectral analysis extracts a canonical ENSO cycle, and provides rectified coordinates representing the cycle as an approximately fixed-speed oscillation. In rectified space it is clear that the representation of the ENSO cycle in terms of Niño indices (SST anomalies) is asymmetric because La Niña appears earlier (in phase/angle space) around the one-dimensional cycle (see Fig. 5(f, j)). Without the rectified representation, it would be difficult to assign a characteristic speed/frequency around the cycle. This notion of characteristic frequency will be useful below for constructing phase composites, and should also be useful for constructing reduced models. More broadly, we suggest that rectification is an important conceptual construction, which should be useful in a wide range of climate dynamics applications.

**ENSO phases and their associated composites.** We construct reduced representations of the ENSO lifecycle by partitioning the 2D phase spaces associated with the generator eigenfunctions and lagged Niño 3.4 index into angular phases, and then study the properties of associated phase composites of relevant oceanic and atmospheric fields. Figure 6(a, b, d, e) depicts the phase space partitions over eight such phases for CCSM4 and ERSSTv4, respectively. Each phase is constructed from samples at times for which $|g_j|$ lies in the top $m$ values in the corresponding 45° radial sector, where $m = 200$ and 20 for CCSM4 and ERSSTv4, respectively. Larger magnitude values of the eigenfunction $g_j$ occur at times belonging to stronger ENSO cycles, and because we seek a strong canonical ENSO cycle, we subsample at these times. Mathematically, the phase composites constructed in this manner can be interpreted as conditional expectations of observables (e.g., SST anomaly fields) with respect to a discrete variable $\pi_j: \Omega \to \{0, 1, \ldots, 8\}$ indexing the eight phases associated with eigenfunction $g_j$. The inclusion of a "zero" phase nominally is to account for states which are not ENSO-active, consistent with earlier work[24,78] that prioritizes larger values of real eigenfunctions and equivariant functions; see Methods for further details.

It should be noted that in the eigenfunction-based representation, partitioning the phase space into phases of uniform angular extent is a natural choice since the evolution is rectified and takes place at an approximately constant angular frequency. In other words, in the eigenfunction picture in Fig. 6(b, e), phases of uniform angular extent correspond to phases of uniform temporal duration, in this case approximately $4/8 = 1/2$ years. In the case of the Niño 3.4-based representation in Fig. 6(a, d), achieving a well-balanced partitioning is more challenging due to variable/retrograde angular speed and significant radial motion. Here, we have opted to employ a uniform partitioning scheme which is common practice with many cyclical climatic indices, including indices for the MJO and other intraseasonal oscillations[79]. We note that this is already an improvement over a characterization of ENSO phases based on scalar indices, since such representation cannot distinguish the time tendency (increasing or decreasing) of the oscillation.

In both the Niño 3.4- and eigenfunction-based representations, the phases are numbered such that Phase 1 corresponds to El Niño, and periodic cycling of the phases from 1 to 8 represents an El Niño to La Niña to El Niño evolution. Turning back to the Niño-3.4 representation in Fig. 5(b), Phase 5 is a La Niña phase centered at angle $\pi$. On the other hand, in the generator representation in Fig. 5(f), La Niña (deep blue, corresponding to lowest Niño-3.4 values) occurs at Phase 4, centered at $3\pi/4$, due to the rectification. This means that the rectified generator representation allocates more phases (Phases 5–8) in the La Niña to El Niño portion of the ENSO lifecycle, thus yielding a more granular description of ENSO initiation processes.

In Fig. 7, we examine phase composites of monthly averaged SST and surface wind anomalies, constructed using the Niño 3.4 and generator phases from CCSM4 and ERSSTv4 depicted in Fig. 7. In the the CCSM4 analysis we use surface wind data from the atmospheric component of the model (CAM2). In the ERSSTv4 analysis, the surface wind data is from the NCEP/NCAR Reanalysis 1 product[80]. First, on a coarse level, both the Niño- and generator-based composites recover the salient features of the ENSO lifecycle. These include (i) the characteristic El Niño "tongue" of positive SST anomalies in the Eastern equatorial Pacific, together with its associated anomalous surface westerlies, in Phase 1; (ii) meridional discharge in the ensuing intermediate phases; and (iii) formation of negative SST anomalies and easterly surface winds during the La Niña phases (Phases 5 and 4 for the Niño- and eigenfunction–based representations, respectively).

The Niño-3.4 and generator-based composites in Fig. 8 also exhibit important differences, particularly in the La Niña to El Niño transition phases. In both CCSM4 and ERSSTv4, Phases 6–8 of the generator capture a reorganization of the large-scale surface winds from a convergent configuration over the Maritime Continent in Phase 6 to a divergent configuration initiating in Phase 7 with a buildup of anomalous westerlies in the Western Pacific, developing further in Phase 8. In particular, the anomalous westerlies in Phase 7 are consistent with the aggregate effect of higher-frequency, stochastic atmospheric variability such as westerly wind bursts[81] that trigger the development of El Niño events.

To examine this behavior in more detail, in Fig. 8 we show phase-composited zonal wind profiles at the dateline for the latitude range 40°S–40°N. These composites recover a number of important atmospheric features of the ENSO lifecycle, including (i) the mature El Niño state in Phase 1 characterized by strong westerlies in the tropics maintaining positive SST anomalies in the eastern part of the Pacific basin; (ii) El Niño decay in Phase 2 with decreasing easterly intensity and a southward shift[82] of the anomalous equatorial westerlies; (iii) La Niña initiation in Phase 3; (iv) La Niña growth, saturation, and decay in Phases 4–6; (v) El Niño initiation in Phase 7, featuring a clear signal of anomalous westerlies; and (vi) El Niño growth in Phase 8, cycling back to the mature El Niño state in Phase 1. These features are resolved in both the CCSM4 and ERSSTv4 datasets, though the observational composites tend to display a higher degree of asymmetry between the Northern and Southern hemispheres.

In contrast to the generator composites, the Niño-3.4-based composites exhibit significantly more abrupt El Niño–La Niña and La Niña–El Niño transitions, failing to recover a number of the processes outlined above. In particular, Niño Phase 2 (which represents El Niño decay in the generator picture), closely resembles the mature El Niño phase in Phase 1. In Phase 3, the Phase 2 configuration is abruptly replaced by near-neutral conditions, failing to capture the southward shift of the anomalous equatorial westerlies associated with El Niño termination. The Niño-based composites are characterized by a similarly abrupt La Niña to El Niño transition in Phases 7 and 8, with weak negative SST

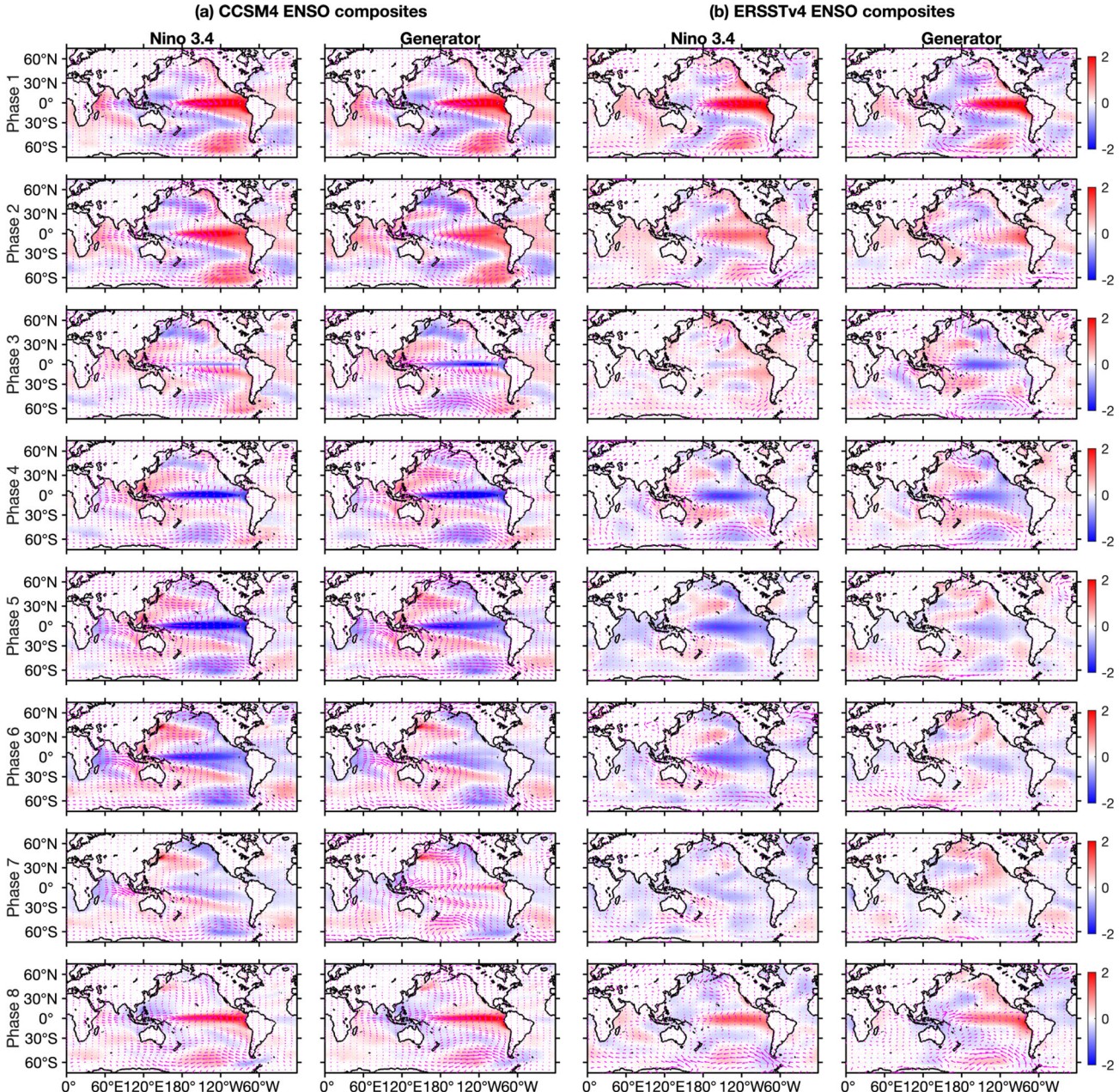

**Fig. 7 Phase composites of the ENSO lifecycle.** The panels show composites of SST anomalies (K; colors) and surface wind (arrows) anomalies from **a** CCSM4 and **b** ERSSTv4 and NCEP/NCAR Reanalysis, based on the Niño 3.4 (first and third columns from left) and generator phases (second and fourth columns) from Fig. 6. Phases advance row-wise from Phase 1 (top row) to Phase 8 (bottom row).

anomalies in the eastern equatorial Pacific being replaced by well-developed El Niño conditions. Importantly, there is no representation of anomalous westerlies during these phases. The more physically informative reconstruction of the ENSO lifecycle provided by the generator is likely due to the dynamical rectification property discussed above, which enables phase partitioning in the "intrinsic" phase of the oscillation. Beyond ENSO, we expect this rectification property to be beneficial in diagnostic and mechanistic studies of different climate phenomena.

**Phase equivariance.** Besides the diagnostic aspects described above, an important requirement of an index representing a coherent oscillatory phenomenon such as ENSO is that phase progression is consistent with the temporal evolution of the samples constituting each phase—this is the concept of phase

equivariance stated in the Introduction. In the particular setting of the eight-phase ENSO reconstruction studied here, phase equivariance means that the forward evolution of the samples that constitute phase $i$ by six months (the nominal duration of each phase) should map these samples into the samples making up phase $i+1$, modulo 8. Theoretically, this correspondence should be exact for a purely periodic process such as the variable-speed oscillator in Fig. 4, but for a chaotic oscillator such as ENSO we expect it to hold only approximately. We will demonstrate below that the indices based on the generator eigenfunctions $g_7$ (CCSM4) and $g_6$ (ERSSTv4) exhibit greater equivariance than the lagged Niño 3.4 index $f_{nino}$.

To test for equivariance in the Niño-3.4 and generator-based representation of ENSO, in Fig. 9 we show this forward evolution in the corresponding 2D phase spaces in six-month increments

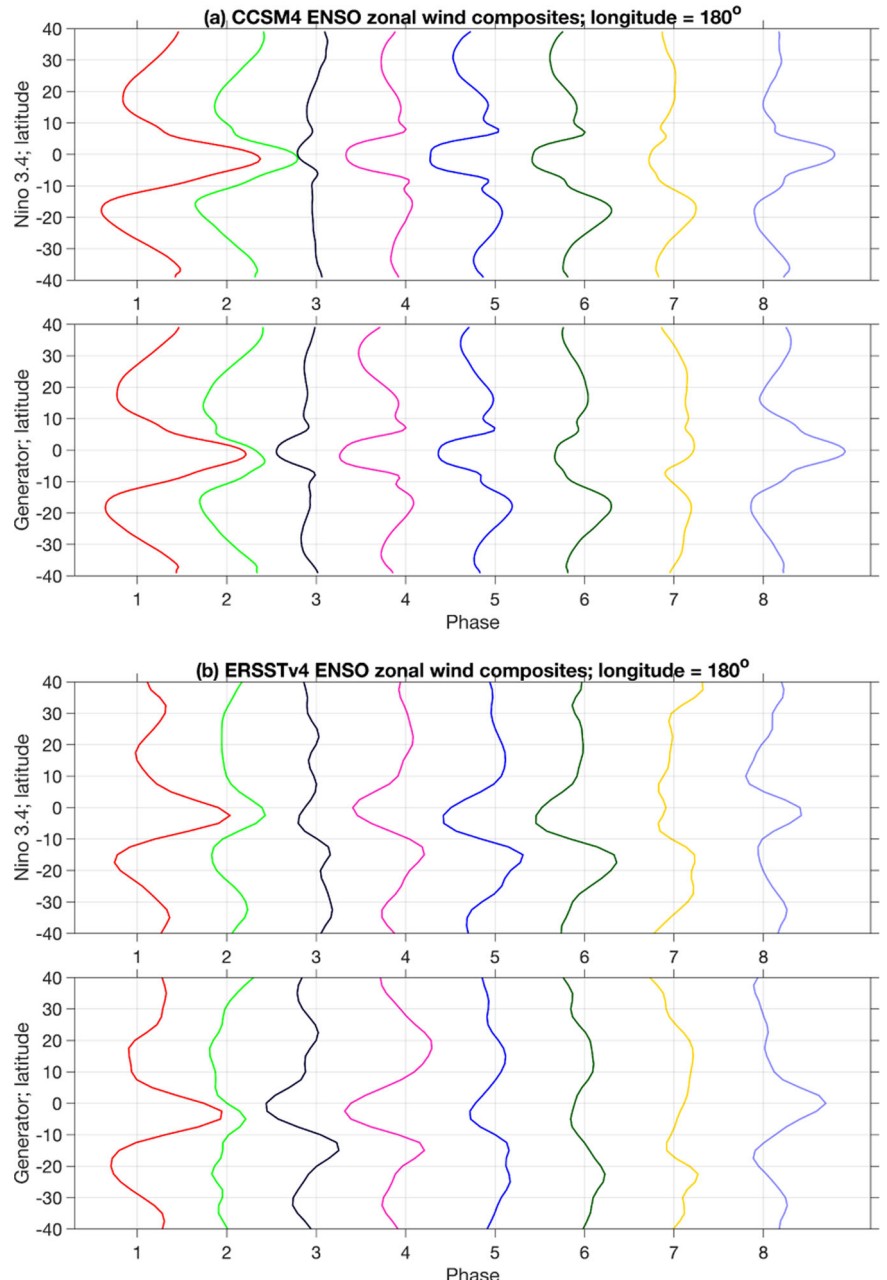

**Fig. 8 Phase-composited surface zonal wind profiles for CCSM4 (a) and ERSSTv4 (b).** Each set of panels shows composites of surface zonal winds sampled at the dateline along the meridional interval 40°S–40°N, based on the Niño 3.4 (top panels) and generator (bottom panels) ENSO phases from Fig. 6.

starting from Phase 7 (i.e., the phase most closely related to El Niño initiation). There, it is evident that the generator lifecycle exhibits phase equivariance on significantly longer intervals than the Niño 3.4 lifecycle, in both CCSM4 and ERSSTv4. In the case of the generator, the centroid of the cloud of points making up the forward evolution of Phase 7 has a phase angle consistent with equivariant phase evolution over the examined 2-year interval. While there is visible dispersion occurring by ≃12 months, this dispersion occurs predominantly in the radial direction and has a limited effect on the phase classification. In contrast, the point clouds corresponding to forward evolution of the Niño-based Phase 7 exhibit strong dispersion in both radial and angular directions, decorrelating with the target phase expected from equivariance on intervals as short as 6–12 months. The difference in equivariance

between the Niño 3.4 and generator lifecycle is most striking in the ERSSTv4 data, where after a 1-year interval the forward-evolved Phase 7 from Niño 3.4 has zero overlap with the expected Phase 1, whereas in the case of the generator that overlap is close to 100%. These results open the possibility that methods of characterizing ENSO based on area-averaged anomalies (such as the lagged Niño 3.4 index $f_{nino}$) may conflate unrelated parts of the cycle. This could contribute to difficulties with ENSO prediction, as shown in Fig. 9(a, b); (top) for $f_{nino}$, where poorly chosen groupings mix together unrelated ENSO phases, leading to rapid divergence of these "false" groupings. Our results suggest that ENSO may have a more significant cyclic component than previously realized.

As a more quantitative assessment of phase equivariance, in Supplementary Fig. 2 we show the fractional sample overlap

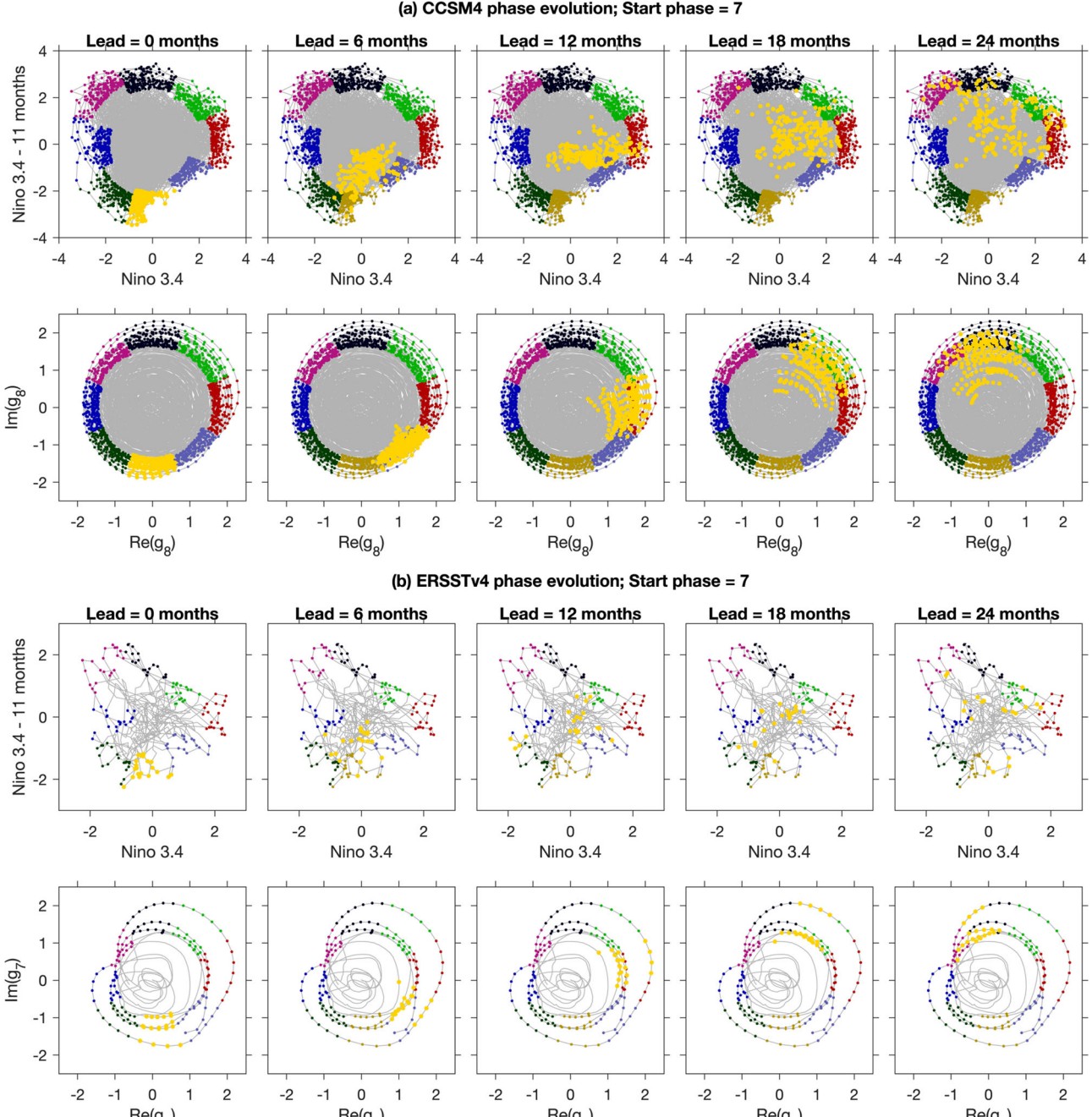

**Fig. 9 Evolution of ENSO Phase 7 at six-month increments for CCSM4 (a) and ERSSTv4 (b).** Evolution of ENSO Phase 7 at six-month increments for **a** CCSM4 and **b** ERSSTv4. In each set of panels, the top and bottom rows depict the phase evolution associated with the lagged Niño 3.4 indices $f_{nino}$ and generator eigenfunctions $g_j$, respectively. Bold yellow dots show the Phase 7 members (left column) and their forward images (right four columns) under the dynamics. Dots colored in muted colors show the phase partitioning from Fig. 6 for reference. Observe that the generator-based evolution undergoes a uniform phase progression with significantly smaller spread than the Niño-based evolution.

between the forward-evolved ENSO phases in CCSM4 in six-month increments with the expected target phases from equivariance. It is worthwhile noting that highest predictability of the generator phases occurs for start phases near the El Niño/La Niña peaks (Phases 1, 2, and 6), where the fractional overlap remains above 0.5 for at least a year. The evolution initialized at intermediate phases such as 3–5, 7, and 8 is somewhat less equivariant, with the relative overlap dropping to smaller than 0.5 values after a year. This behavior may be a manifestation of the ENSO spring predictability barrier[83].

**ENSO diversity**. ENSO diversity, i.e., the tendency of El Niño/La Niña events to differ from each other in terms of their spatial and temporal characteristics, has been a topic of considerable interest in the literature[4,84–87]. It is common to spatially classify El Niño events as being of Eastern Pacific (EP) or Central Pacific (CP) type, depending on the longitudinal location of the highest SST anomalies[4]. Some studies have interpreted these patterns as being the outcome of distinct temporal processes, with CP events dominated by quasi-biennial (QB; 1.5–3 yr) components, and strong EP events exhibiting both QB and low-frequency (LF)

components in the 3–7 yr band[87]. Other studies have classified ENSO events as cyclic, episodic, or multiyear, depending on whether they are preceded by the opposite, neutral, or same phase, respectively[86]. In this section, we show how the generator eigenfunctions extracted from ERSSTv4 can account for some inter-event differences in period 1975–2020. That period saw the occurrence of three strong EP El Niños (1982/83, 1997/98, and 2015/16), one moderate EP El Niño (1986/87), two CP El Niños (1994/95 and 2009/10), and two events which were of mixed character (1991/92, 2002/03)[4].

Recall from Fig. 3(b) that the top part of the generator spectrum exhibits the fundamental ENSO eigenfunctions (with a 4 yr eigenperiod), the associated ENSO combination modes (with various eigenperiods in the interannual to seasonal band), and also a pair of eigenfunctions, $g_{19}$ and $g_{20}$, with $a \simeq 3$ yr eigenperiod (not shown in Fig. 3(b); see Supplementary Table 3). To assess the contribution of these eigenfunctions in the variability of the Niño 3.4 index, we compute associated time series reconstructions, or "modes", using the standard approach employed in SSA, EEOF analysis, and other comparable techniques utilizing delay embedding[55,69]. Given a complex-conjugate pair of generator eigenfunctions, $\{g_j, g_{j+1}\}$, this procedure produces a (real) time series that represents the component of the Niño 3.4 index reconstructed by the pair $\{g_j, g_{j+1}\}$. Moreover, the time series from several such pairs can be added together to produce reconstructions of Niño 3.4 based on groups of generator eigenfunctions. (See Methods for details of the reconstruction procedure.) In Fig. 10(a), we present reconstructed Niño 3.4 time series based on the fundamental 4-year ENSO mode (red line), the 3-year ENSO mode (blue line), and the sum of the leading two ENSO combination modes (green line). The sum of these ENSO-related modes is also shown (orange line), and captures greater variability than the fundamental ENSO mode alone. Shaded time intervals indicate periods where the the 2-month running correlation coefficient between the latter reconstruction and the Niño 3.4 index is greater than 0.9.

First, it is readily apparent that certain El Niños are well captured by a small number of leading modes—i.e., modes that reflect greater dynamical persistence and cyclicity. In particular, consider the very intense 1982/83 and 1997/98 El Niños: for these two events, the peaks of all three ENSO-related modes are effectively coincident in Fig. 10(a). The next most intense event, 2015/16, is characterized by 4-year mode amplitude comparable to 1982/83, with overall correlation among the 4-year, 3-year, and combination modes. However, the 3-year ENSO mode for 2015/16 is less intense than the corresponding mode for 1982/3.

Consider now the 1986/87 event, which is shown in detail in Fig. 10(b). In this case, we see a distinct behavior, as the combination modes have two peaks, one occurring before and one after the peak of the 4-year mode, and a trough occurring during the peak of the 4-year mode (see green and red lines in Fig. 10(a)). Superposing the combination modes with the fundamental ENSO mode (green line in Fig. 10(b)) results in consecutive peaks in the reconstructed Niño 3.4 index around the peak of the 4-year mode. If we additionally include the 3-year mode (orange line) the relative amplitude of the two peaks changes. In contrast, if we superpose only the 3-year and 4-year modes, the two consecutive peaks do not occur (see blue line in Fig. 10(b)).

Previously, ENSO combination modes have received significant attention due to their role in El Niño termination in boreal spring[65,82,88]. The results in Fig. 10 show that ENSO combination modes can also play an important role in reconstructing events with multiple peaks. We note that while we have directly computed the combination eigenfunctions, in theory (as discussed in subsection "Eigenvalue frequency analysis of monthly-averaged Indo-Pacific

SST") they may be determined from the state of the annual and the 4-year ENSO eigenfunction. Thus, our results show that certain doubly-peaked events, such as the 1986/87 El Niño, can be reconstructed from the same small set of modes as those used to reconstruct strong EP events.

It is noteworthy that the 1982/83, 1986/87, and 1997/98 El Niños, as well as the 2009/10 event (which is also well captured by the reconstructions in Fig. 10(a)), are all followed by La Niñas in the subsequent year. In the transition-based classification of ENSO[86], these La Niñas are thus all classified as cyclic (cyclic El Niños are defined in a symmetric way). From the perspective of our spectral analysis approach, the occurrence of these cyclic La Niñas can be explained from the fact that once a generator eigenfunction $g_j$ becomes "active", i.e., $|g_j(\omega)|$ is large for a given climate state $\omega$, it will, with high likelihood, remain active for at least a significant fraction of the cycle that it represents (since $g_j(\omega)$ precesses in the complex plane with fixed frequency and a weaker radial motion; see, e.g., Fig. 6(e)). In particular, significant El Niño events in the generator-based representation have high likelihood of leading to La Niñas in the following year. On the other hand, the generator eigenfunctions have only moderate magnitude in the La Niña phase (see Fig. 5(m, n)). This suggests fewer cyclic El Niños, which is consistent with the different triggering mechanisms of the two phenomena[86].

In contrast to all of the events mentioned above, other events such as the 1991/92 El Niño, are not readily accounted for by the leading modes. It is possible that this event is tied to the June 1991 eruption of Mt. Pinatubo[89–91]; external forcing of this event may explain why the eigenmodes fail to capture it. We note that the 1982/83 El Niño followed the eruption of El Chichón, but is nonetheless a strong EP event captured by the leading ENSO modes. The 1994/95 CP El Niño represents an example of another "missed" event in the context of the modes illustrated in Fig. 10. The fact that the leading generator eigenfunctions, which favor cyclicity and dynamical persistence, do not capture these events is consistent with the broadly accepted observation that CP events exhibit less canonical behavior than their EP counterparts[4]. Intriguingly, the reconstructions in Fig. 10 indicate the existence of a 3-year interannual mode, associated with generator eigenfunctions $g_{19}$ and $g_{20}$, which plays a significant role in strong EP events but does not significantly contribute to CP events. This differs somewhat from previous mode decompositions of the Niño 3.4 index[87], which have identified a single QB mode contributing to both CP and strong EP events.

In summary, the results in Fig. 10 show that our spectral approach can differentiate certain ENSO events in terms of amplitude and phasing of an underlying set of dominant modes. What we would argue here, at least from the viewpoint afforded by a single regional index like Niño-3.4, is that a rich diversity of El Niño behavior can be "constructed" from a small number of eigenfunctions of a dynamical operator. That the ENSO combination modes also contribute in a discernible way, either by adding to the fundamental and 3-year ENSO modes as in 1982/83 or 1997/98 or creating consecutive peaks as in 1986/87, is also of note, as these modes may not be separately identified in the variance basis of EOFs, although they have been noted in SSA.

## Discussion

Operator-theoretic approaches for dynamical systems, realized through kernel methods for machine learning, provide an effective framework for identification of persistent cyclic modes of variability in climate dynamics. Central to this framework is modeling the evolution of observables of the climate system with transfer and Koopman operators. The dominant eigenfunctions of these operators yield succinct and physically interpretable

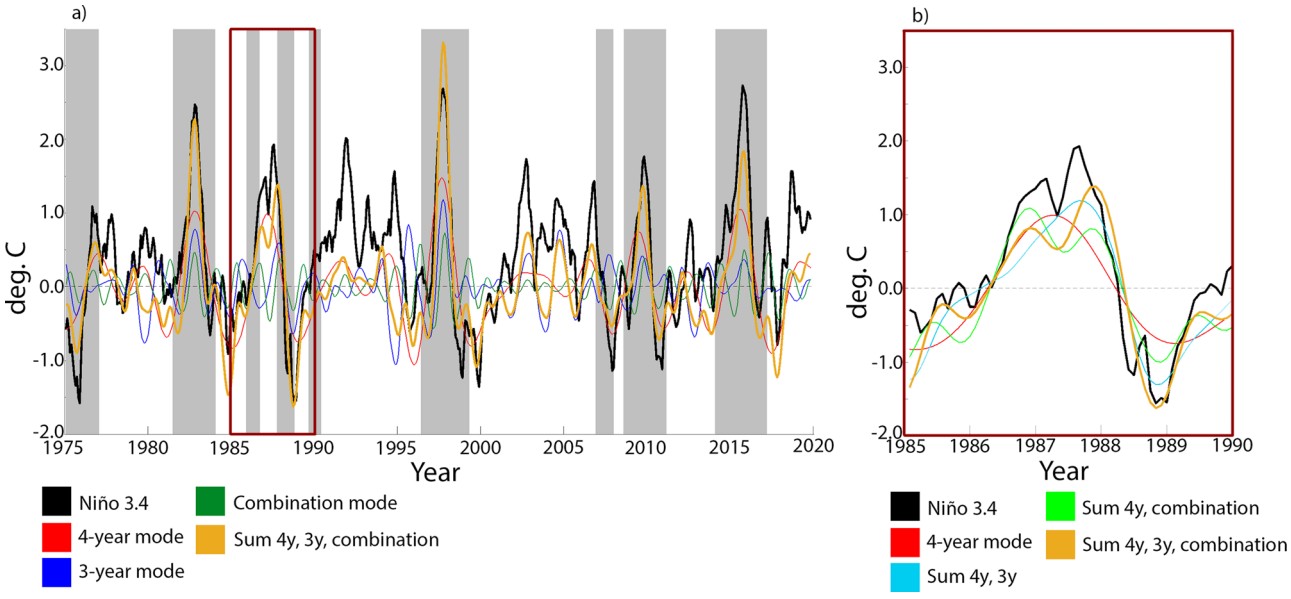

**Fig. 10 Raw (solid black line) and reconstructed (colored lines) Niño 3.4 index for the ERSSTv4 data.** The reconstructions are based on groups of eigenfunctions of the generator from Fig. 3 and Supplementary Table 3. See Methods for a description of the reconstruction procedure. **a** Reconstructions over the period January 1975 to February 2020 based on (i) the fundamental (4-year) ENSO pair, $g_5, g_6$ (red); (ii) leading two ENSO combination pairs, $g_{10}, g_{11}, g_{15}, g_{16}$ (green); (iii) 3-year ENSO pair, $g_{19}, g_{20}$ (blue); and (iv) the sum of the modes in (i–iii) (orange). The eigenfunction index sets $J$ (see Methods) employed for these reconstructions are (i) {5, 6}; (ii) {10, 11, 15, 16}; (iii) {19, 20}; and (iv) {5, 6, 10, 11, 15, 16, 19, 20}. Time intervals (shaded gray) indicate periods where the running correlation coefficient between the Niño 3.4 index and the reconstructed index based on all modes (Case (iv); orange line) exceeds 0.9. The running correlation coefficient was computed using a two-month (centered) sliding window. **b** A detailed view of the "double-peaked" 1986/87 El Niño, highlighting the role of the combination modes (see green line) in reconstructing the double peak of the Niño 3.4 index. The time interval depicted in **b** is indicated by a red box in **a**.

representations of fundamental modes of climate variability, with the corresponding eigenvalues reflecting the intrinsic timescale of variability of the mode. We have shown by means of theoretical arguments and numerical analyses of (i) idealized dynamical systems, (ii) comprehensive climate models, and (iii) reanalysis data, that these eigenfunctions reveal approximate cycles embedded in complicated systems with several advantageous characteristics over conventional approaches. Composites in the original observation space can be readily constructed; see Fig. 1. A further distinguishing aspect of our eigenfunctions is that they provide rectified coordinates for the state of the oscillation (Figs. 5 and 6), making them better suited for indexing the fundamental oscillations of the climate. Moreover our extracted cycles display a high level of self-consistency under forward evolution (Fig. 9), a desirable property for characterizing a canonical strong ENSO and promising for prediction.

A major focus of this work has been the El Niño Southern Oscillation, extracted from monthly-averaged Indo-Pacific SST data from a millennial control integration of a comprehensive climate model (CCSM4) and reanalysis data (ERSSTv4). In both of these datasets, the generator spectrum (Fig. 3) contains a pair of slowly decaying eigenfunctions with an interannual eigenfrequency, providing a rectified representation of the canonical ENSO lifecycle. In addition to the fundamental ENSO modes, the spectrum of the generator is found to exhibit a hierarchy of combination modes between ENSO and the annual cycle with the theoretically expected frequencies. These combination modes appear to play a role in capturing "double El Niño" events in the recent observational record, such as the 1986/87 series of events. Meanwhile, other events, such as the 1991/92 El Niño following the Mt. Pinatubo eruption and the 1994/95 central Pacific El Niño are not captured by the leading eigenfunctions, suggesting a different dynamical origin. Going beyond cyclic behavior, in the case of the reanalysis data, the spectrum of the generator was found to

contain nonstationary modes associated with climate change, as well as combination modes representing the modulation of the annual cycle by the climate-change trend. Our analysis motivates further application of the spectral theory of dynamical systems to diagnosing and predicting the fundamental dynamical patterns of the climate.

## Methods

As described in the Results, we have a time-ordered dataset $x_1, \ldots, x_N \in \mathbb{R}^d$, arising as a series of observations $x_n = X(\omega_n)$ from a trajectory of an abstract dynamical system $\Phi^t: \Omega \to \Omega$, where $\omega_n = \Phi^{n\,\Delta t}(\omega_0)$. In our experiments, the SST field sampled at $d \gg 1$ Indo-Pacific gridpoints at time-index $i$ yields a vector $x_i \in \mathbb{R}^d$ (see Supplementary Table 1 for further details on the datasets employed in this study). We also consider low-dimensional examples with $d = 3$ (L63 system; Fig. 2, Supplementary Table 2) and $d = 2$ (variable-frequency oscillator; Fig. 4), where $X$ is the identity map on the respective state space $\Omega$. Recall that $\Delta t > 0$ is the sampling interval, and $\mu$ is an assumed physically meaningful invariant probability measure for $\Phi^t$, $\mu = \mu \circ \Phi^{-t}$. In what follows, we describe data-driven techniques for approximation of (i) the transfer operator $P^{\Delta t}$, or the Koopman operator $U^{\Delta t}$; and (ii) the generator $V$ of the transfer/Koopman operator semigroups. In the measure-preserving setting, the transfer and Koopman operators on $H = L^2(\Omega, \mu)$ form dual pairs related by adjoints, $(P^t)^* = U^t$ for every $t \in \mathbb{R}$. Thus, for conciseness of exposition, in what follows we focus on approximation of the transfer operator $P = P^{\Delta t}$, corresponding to $\Phi = \Phi^{\Delta t}$. In addition, we describe our procedure for computing spatiotemporal mode reconstructions from eigenfunctions.

**Delay embedding**. We will delay-embed the data to form vectors $\tilde{x}_i = (x_{i-(Q-1)\ell}, \ldots, x_{i-\ell}, x_i) \in \mathbb{R}^{Qd}$ for some positive integers $Q$ and $\ell$, to have an improved estimation of the underlying state $\omega_i \in \Omega$ as in standard Takens embedding[50,51]. Let $\nu$ be the measure induced on $\mathbb{R}^{Qd}$ by the invariant measure $\mu$ on $\Omega$. We seek to approximate projected versions of the operators $P^{\Delta t}$, $U^{\Delta t}$, and $V$ that act on functions on a space of projected observables $L^2(\mathbb{R}^{Qd}, \nu)$. In practice, integrals with respect to $\nu$ are approximated by integrals with respect to the sampling probability measure $\nu_N := \sum_{i=(Q-1)\ell}^{N-1} \delta_{\tilde{x}_i}/(N - (Q-1)\ell)$, where $\delta_{\tilde{x}_i}$ is the Dirac $\delta$-measure centered at $\tilde{x}_i$. This will shortly reduce to summing over the original data points $\tilde{x}_i$.

**Approximation of transfer and Koopman operators**. We define a novel, data-driven Markov chain approximation of $P$, where each embedded data point $\tilde{x}_i \in \mathbb{R}^{Qd}$, $i = (Q-1)\ell, \ldots, N-2, N-1$ is identified with a Markov state. Our approximation retains important structural properties of $P$, namely it is a positive operator (nonnegative functions are mapped to nonnegative functions) and preserves integrals with respect to the data-based measure $\nu_N$. The addition of noise mentioned in the main text to form $P_\epsilon$ is done via Gaussian kernels

$$k_\epsilon(\tilde{x}_i, y) = \exp\left(-\|\tilde{x}_i - y\|^2/\epsilon^2\right), \qquad \tilde{x}_i, y \in \mathbb{R}^{Qd}, \qquad (1)$$

centered on each point $\tilde{x}_i$, where $\epsilon$ is a positive bandwidth parameter, the choice of which is discussed at the end of this subsection.

We discretely approximate the Markov operator $P_\epsilon : L^2(\mathbb{R}^{Qd}, \nu) \to L^2(\mathbb{R}^{Qd}, \nu)$ defined by

$$P_\epsilon f(z) = \int_{\mathbb{R}^{Qd}} \left(\frac{k_\epsilon(z, \Phi(y))}{\int_{\mathbb{R}^{Qd}} k_\epsilon(z', \Phi(y)) \, d\nu(z')}\right) f(y) \, d\nu(y)$$

as

$$P_\epsilon f(z) \approx \int_{\mathbb{R}^{Qd}} \left(\frac{k_\epsilon(z, \Phi(y))}{\int_{\mathbb{R}^{Qd}} k_\epsilon(z', \Phi(y)) \, d\nu_N(z')}\right) f(y) \, d\nu_N(y).$$

Evaluating $P_\epsilon f$ at an embedded data point $\tilde{x}_i$, we have

$$\begin{aligned}
P_\epsilon f(\tilde{x}_i) &\approx \int_{\mathbb{R}^{Qd}} \left(\frac{k_\epsilon(\tilde{x}_i, \Phi(y))}{\int_{\mathbb{R}^{Qd}} k_\epsilon(z', \Phi(y)) \, d\nu_N(z')}\right) f(y) \, d\nu_N(y) \\
&= \sum_{j=(Q-1)\ell}^{N-2} \left(\frac{k_\epsilon(\tilde{x}_i, \tilde{x}_{j+1})}{\sum_{i'=(Q-1)\ell}^{N-2} k_\epsilon(\tilde{x}_{i'}, \tilde{x}_{j+1})}\right) f(\tilde{x}_j) \\
&= \sum_{j=(Q-1)\ell}^{N-2} P_{ij} f(\tilde{x}_j),
\end{aligned}$$

where

$$P_{ij} = \frac{k_\epsilon(\tilde{x}_i, \tilde{x}_{j+1})}{\sum_{i'=(Q-1)\ell}^{N-2} k_\epsilon(\tilde{x}_{i'}, \tilde{x}_{j+1})}.$$

The matrix $\boldsymbol{P} = [P_{ij}]$ is column stochastic and we may think of $\boldsymbol{P}_{ij}$ as the conditional probability that the state $j$ (or data point $\tilde{x}_j$) transitions to the state $i$ (or data point $\tilde{x}_i$), in one time step, according to the kernel $k_\epsilon$ and the measure $\nu_N$. A function $f : \mathbb{R}^{Qd} \to \mathbb{C}$ taking values $\boldsymbol{f}_0 := f(\tilde{x}_i)$, $i = (Q-1)\ell, \ldots, N-2$, is evolved forward in time by matrix-vector multiplication $\boldsymbol{Pf}$; this approximates the action of $P_\epsilon f$. By construction, the mass conservation property $P_\epsilon^* \mathbf{1} = \mathbf{1}$ is inherited by $\boldsymbol{P}$, namely $\boldsymbol{P}^\top \mathbf{1} = \mathbf{1}$.

In practice, we compute the numerical spectrum of $\boldsymbol{P}$, $\boldsymbol{Pg}_j = \Lambda_j \boldsymbol{g}_j$, and extract the most persistent cyclic behavior from the eigenvector $\boldsymbol{g}_1$ corresponding to the eigenvalue $\Lambda_1 = re^{i\alpha}$ with largest magnitude inside the unit circle (largest $|r| < 1$) and $\alpha > 0$. As described in the Results, $\alpha$ represents the angle of rotation around the extracted cycle per unit time. The corresponding eigenvector $\boldsymbol{g}_1$ approximates the eigenfunction $g^{(\epsilon)}(\tilde{x}_i)$ of $P_\epsilon$ that approximately projects the system from $\mathbb{R}^{Qd}$ to the most persistent cycle (approximately lying on $S^1$) as illustrated, e.g., in Fig. 2(f–h) in the context of the L63 system.

In the experiment in Fig. 2(f–h), we used $N = 16{,}000$ samples taken at a sampling interval of $\Delta t = 0.01$ time units; see Supplementary Table 2. Moreover, the dimension is $d = 3$, and we do not need to embed the data as we have access to the original state, thus we set $Q = 1$. In calculations using the ERSSTv4 ($N = 600$) and CCSM ($N = 15{,}600$) Indo-Pacific SST datasets, using $\Delta t = 1$ month we find that a single lag of $Q = 2$ with $\ell = 12$ months (approximately one quarter of the cycle period) is sufficient to accurately extract an ENSO frequency and ENSO eigenfunctions; see Supplementary Table 1.

Regarding the choice of $\epsilon$, generally one wishes to select an $\epsilon$ as small as possible, while maintaining an eigenvalue 1 of $\boldsymbol{P}$ with unit multiplicity. A ballpark estimate of suitable $\epsilon$ is the mean nearest neighbor distance (averaged over all embedded data points $\tilde{x}_i$), divided by $\sqrt{2}$; this scales the Gaussian in (1) to have greatest slope (and therefore "distinguishing ability") when $\|\tilde{x}_i - y\|$ is the mean nearest neighbor distance. The values of $\epsilon$ used in the Lorenz, ERSSTv4, and CCSM calculations are shown in Supplementary Tables 1 and 2, and are modified by less than a factor of four from the above ballpark estimate.

**Approximation of the generator**. Our method outputs a collection of $\tilde{N}$-dimensional complex vectors $\boldsymbol{g}_0, \ldots, \boldsymbol{g}_L \in \mathbb{C}^{\tilde{N}}$, with $\tilde{N} = N - Q + 1$, $\boldsymbol{g}_j = (g_{(Q-1)j}, \ldots, g_{(N-1)j})^\top$, and complex numbers $\hat{\lambda}_0, \ldots, \hat{\lambda}_L$, such that $g_{ij}$ approximates the value of an eigenfunction $g_j^{(\epsilon)}$ of the regularized generator $V_\epsilon$ at state $\omega_i \in \Omega$, and $\hat{\lambda}_j$ approximates the corresponding eigenvalue, $\lambda_j$. That is, we have $g_{ij} \approx g_j(\omega_i)$ and $\hat{\lambda}_j \approx \lambda_j$, where $V_\epsilon g_j = \lambda_j g_j$. The numerical procedure to compute the eigenpairs $(\hat{\lambda}_j, \boldsymbol{g}_j)$ consists of two parts:

1. Computation of basis vectors $\boldsymbol{\phi}_0, \ldots, \boldsymbol{\phi}_{L-1} \in \mathbb{R}^{\tilde{N}}$ as eigenvectors of an $\tilde{N} \times \tilde{N}$ kernel matrix $\tilde{\boldsymbol{K}}$ constructed from the data.
2. Formation of an $L \times L$ matrix $\boldsymbol{W}$ approximating the operator $V_\epsilon$ and solution of the associated eigenvalue problem.

In what follows, we outline these steps, referring the reader to our previous work[34] for additional details and pseudocode.

*Kernel matrix and basis functions.* Using the delay-embedded data $\tilde{x}_i$, we compute an $\tilde{N} \times \tilde{N}$ matrix $\boldsymbol{K}$, whose entries are given by the values $K_{ij} = k_\gamma(\tilde{x}_i, \tilde{x}_j)$ of a pairwise kernel function $k_\gamma : \mathbb{R}^{dQ} \times \mathbb{R}^{dQ} \to \mathbb{R}$. We use variable-bandwidth kernels

$$k_\gamma(\tilde{x}_i, y) = \exp\left(-\frac{\|\tilde{x}_i - y\|^2}{\gamma^2 \sigma^2(\tilde{x}_i, y)}\right), \qquad \tilde{x}_i, y \in \mathbb{R}^{Qd}, \qquad (2)$$

centered on each point $\tilde{x}_i$, where $\gamma$ is a positive bandwidth parameter, and $\sigma(\tilde{x}_i, y)$ is a positive bandwidth function. Intuitively, the role of $\sigma$ is to control the rate of decay (locality) of the kernel in data-dependent manner, such that in regions of high sampling density $\sigma$ is small, leading to a tighter kernel $k_\gamma$, and allowing resolution of finer-scale features. Conversely, in low-density regions $\sigma$ is large, and hence we obtain a broader kernel $k_\gamma$, enhancing robustness to statistical sampling errors. Note that the radial Gaussian kernel in (1) is a special case of (2) with the constant bandwidth function $\sigma(\tilde{x}_i, y) = 1$ and bandwidth parameter $\gamma = \epsilon$. Here, we use the symbol $\gamma$ for the kernel bandwidth parameter to distinguish it from $\epsilon$ employed for transfer/Koopman operator approximation in the previous section. The choice of $\gamma$ and bandwidth function $\sigma$ will be discussed in a subsequent section. It should be noted that in addition to improving state estimation, delay embedding also improves the efficiency of basis vectors derived from the data $\tilde{x}_j$ in approximating transfer/Koopman operator eigenfunctions[33] (as noted in the main text).

Having constructed the kernel matrix $\boldsymbol{K}$, we next normalize it to obtain a bistochastic kernel matrix, i.e., a symmetric $\tilde{N} \times \tilde{N}$ matrix $\bar{\boldsymbol{K}}$ with positive entries $\bar{K}_{ij}$, satisfying $\sum_{j=Q-1}^{N-1} \bar{K}_{ij} = 1$ for all $i \in \{Q-1, \ldots, N-1\}$. The normalization procedure[92] employs the steps

$$\bar{\boldsymbol{K}} = \hat{\boldsymbol{K}}\hat{\boldsymbol{K}}^T, \qquad \hat{\boldsymbol{K}} = \boldsymbol{D}^{-1}\boldsymbol{K}\boldsymbol{S}^{-1/2},$$

where $\boldsymbol{D}$ and $\boldsymbol{S}$ are diagonal matrices with diagonal entries $D_{ii} = \sum_{j=Q-1}^{N-1} K_{ij}$ and $S_{ii} = \sum_{j=Q-1}^{N-1} K_{ij}/D_{jj}$, respectively. The basis vectors $\boldsymbol{\phi}_j$ are then obtained by solving the matrix eigenvalue problem

$$\bar{\boldsymbol{K}}\boldsymbol{\phi}_j = \eta_j \boldsymbol{\phi}_j, \qquad \eta_j \in [0, 1], \qquad \boldsymbol{\phi}_j = (\phi_{(Q-1)j}, \ldots, \phi_{(N-1)j})^\top.$$

By convention, we order the eigenvalues $\eta_j$ in decreasing order, $\eta_0 \geq \eta_1 \geq \cdots$, and normalize the corresponding eigenvectors such that $\boldsymbol{\phi}_i^\top \boldsymbol{\phi}_j = \tilde{N}\delta_{ij}$. By Markovianity of $\bar{\boldsymbol{K}}$ and strict positivity of $k_\gamma$ (which implies that the elements of $\bar{\boldsymbol{K}}$ are strictly positive), the leading eigenvalue $\eta_0$ is equal to 1, and is strictly greater than $\eta_1$. Moreover, the corresponding eigenvector $\boldsymbol{\phi}_0$ has constant elements, which can be set to 1 by our choice of normalization. As a result, viewed as temporal patterns $t_i \mapsto \phi_{ij}$, the eigenvectors $\boldsymbol{\phi}_j$ with $j > 1$ have zero mean (since they are orthogonal to $\boldsymbol{\phi}_0$) and unit variance (since $\|\boldsymbol{\phi}_j\|^2/\tilde{N} = 1$).

Note that because $k_\gamma$ from (1) is a nonlinear kernel, the entries of $\boldsymbol{\phi}_j$ are not necessarily linear projections of the data $\tilde{x}_i$ onto a corresponding extended EOF (EEOF); that is, in general, $\phi_{ij}$ is not equal to $u_j^\top \tilde{x}_i$ for an EEOF $u_j \in \mathbb{R}^{dQ}$. The $\boldsymbol{\phi}_j$ can therefore be viewed as nonlinear principal components, which are able to span richer spaces of observables than conventional EEOF techniques utilizing linear (covariance) kernels. This property is particularly important for our purposes, since in what follows we will use the $\boldsymbol{\phi}_j$ to build Galerkin approximation spaces for the generator that can act on nonlinear functions.

In what follows, our approach is to fix $L \ll N$ and employ the leading eigenvectors $\boldsymbol{\phi}_0, \ldots, \boldsymbol{\phi}_{L-1}$ as basis vectors for approximating the generator. We choose an $L$-dimensional approximation space with high regularity, which reduces the sensitivity of our generator approximations to sampling errors.

*Spectral analysis of the generator.* Viewing vectors $\boldsymbol{f} = (f_{Q-1}, \ldots, f_{N-1})^\top \in \mathbb{C}^{\tilde{N}}$ as complex-valued temporal patterns $t_i \mapsto f_i$ sampled discretely in time at the sampling interval $\Delta t$, we approximate the generator $V$ by a finite-difference operator $\mathbb{V} : \mathbb{C}^{\tilde{N}} \to \mathbb{C}^{\tilde{N}}$. As a concrete example, used in all generator calculations in this paper, the following is a fourth-order central scheme:

$$(\mathbb{V}\boldsymbol{f})_i = \begin{cases} 0, & Q-1 \leq i < Q+1, \\ \frac{1}{\Delta t}\left(\frac{1}{12}f_{i-2} - \frac{2}{3}f_{i-1} + \frac{2}{3}f_{i+1} - \frac{1}{12}f_{i+2}\right), & 2 \leq i < N-2, \\ 0, & N-2 \leq i < N. \end{cases} \qquad (3)$$

Using (3), we approximate the generator $V$ by the $L \times L$ antisymmetric matrix $\boldsymbol{V}$ with elements

$$V_{ij} = (\tilde{V}_{ij} - \tilde{V}_{ji})/2, \qquad \tilde{V}_{ij} = \boldsymbol{\phi}_i^\top \mathbb{V}\boldsymbol{\phi}_j/\tilde{N}.$$

It can be shown that $\boldsymbol{V}$ provides a data-driven Galerkin approximation matrix for $V$, which converges in a suitable large-data limit[33,34]. Similarly, we construct an $L \times L$ matrix $\boldsymbol{W}$ approximating the diffusion-regularized generator $V_\epsilon = V - \epsilon\Delta$, by defining

$$\boldsymbol{W} = \boldsymbol{V} - \epsilon\boldsymbol{\Delta}.$$

Here, $\Delta$ is a diffusion operator on the Hilbert space of observables $H$, and $\mathbf{\Delta}$ a positive-semidefinite, self-adjoint matrix approximating $\Delta$. We set $\mathbf{\Delta}$ to the diagonal matrix with entries

$$\Delta_{ii} = \frac{E_i}{E_1}, \quad E_i = \frac{1}{\eta_i} - 1. \tag{4}$$

With these definitions in place, and a choice of regularization parameter $\varepsilon > 0$, we solve the $L \times L$ matrix eigenvalue problem

$$\boldsymbol{W}\boldsymbol{u}_j = \hat{\lambda}_j \boldsymbol{u}_j. \tag{5}$$

The eigenvalues $\hat{\lambda}_j$ provide approximations to the eigenvalues $\lambda_j$ of $V_\varepsilon$. Moreover, the eigenvectors $\boldsymbol{u}_j = (u_{0j}, \ldots, u_{(L-1)j})^\top \in \mathbb{C}^L$ contain the expansion coefficients of the approximate generator eigenfunction $\boldsymbol{g}_j$ in the $\boldsymbol{\phi}_i$ basis; that is,

$$\boldsymbol{g}_j = \sum_{i=0}^{L-1} u_{ij} \boldsymbol{\phi}_i.$$

In analogy to the discrete-time case, we order the eigenpairs $(\hat{\lambda}_j, \boldsymbol{g}_j)$ in decreasing order of $\mathrm{Re}\lambda_j$. We normalize the $\boldsymbol{g}_j$ such that $\boldsymbol{g}_j^\dagger \boldsymbol{g}_j = \tilde{N}$, where $^\dagger$ denotes the complex-conjugate transpose. Note that, in general, the eigenvectors $\boldsymbol{g}_j$ are not orthogonal (though they are approximately orthogonal for sufficiently small $\varepsilon$).

The imaginary parts of the eigenvalues, $\mathrm{Im}\hat{\lambda}_j$, represent the angular frequencies (radians per unit time) corresponding to the eigenfunctions $\boldsymbol{g}_j$. In the main text (e.g., Fig. 3) we show the frequencies $\nu_j = \mathrm{Im}\hat{\lambda}_j/(2\pi)$ measuring cycles per unit time. Meanwhile, the real part $\mathrm{Re}\hat{\lambda}_j$ measures the (negative) decay rate of $\boldsymbol{g}_j$ under the evolution semigroup generated by $\boldsymbol{W}$. By construction, $\boldsymbol{W}$ has a constant eigenvector $\boldsymbol{g}_0 = \mathbf{1}$ corresponding to the eigenvalue $\hat{\lambda}_0 = 0$ (i.e., zero decay rate and oscillatory frequency). All other eigenvalues have strictly negative real part, and we order them in order of decreasing $\mathrm{Re}\hat{\lambda}_j$ (i.e., in order of increasing decay rate) by convention.

In separate calculations with synthetic periodic data, we have verified that the $\simeq 17\%$ approximation error of the triennial eigenfrequency in Fig. 3 can be reduced to $\simeq 5\%$ by using a eighth-order finite-difference approximation scheme at a fixed monthly sampling interval $\Delta t$. Since our focus in this work is on lower frequencies (e.g., the interannual ENSO frequency), we have elected to work with the fourth-order scheme in (3), which provides adequate accuracy for the eigenfrequencies of interest while being less sensitive to numerical perturbations than higher-order schemes.

*Bandwidth function and parameter tuning.* In the CCSM4 and ERSSTv4 analyses, we employ a non-separable bandwidth function that promotes connectivity between datapoints whose relative displacement vector is aligned with the local dynamical flow[93], viz.

$$\frac{1}{\sigma(\tilde{x}_i, \tilde{x}_j)} = \sqrt{(1 - \zeta \cos \theta_i)(1 - \zeta \cos \theta_j)},$$

$$\cos \theta_i = \frac{v_i^T(\tilde{x}_j - x_i)}{\|v_i\| \|\tilde{x}_i - \tilde{x}_j\|}, \quad \cos \theta_j = \frac{v_j^T(\tilde{x}_i - x_j)}{\|v_j\| \|\tilde{x}_j - \tilde{x}_i\|},$$

where $0 \leq \zeta < 1$, and $v_i = \tilde{x}_i - \tilde{x}_{i-1}$, $v_j = \tilde{x}_j - \tilde{x}_{j-1}$ are (trajectory tangent) vectors representing the local time tendency of the data. Following Refs [69–71], we set $\zeta$ to a value close to 1, namely $\zeta = 0.995$ (see Supplementary Table 1). This has the effect of promoting slow timescales in the extracted basis functions $\phi_j$, which reduces the error of the finite-difference approximation of the generator.

Using the dataset and parameters in Supplementary Table 2, we have computed approximate generator eigenfunctions for the L63 system, which exhibit persistent cyclicity analogously to the transfer operator experiment in Fig. 2. In these experiments, we employ a separable bandwidth function[40],

$$\sigma^2(\tilde{x}_i, y) = (\rho(\tilde{x}_i)\rho(y))^{-1/m}, \tag{6}$$

where $\rho(\tilde{x}_i)$ is an estimate of the sampling density of the data at $\tilde{x}_i$, and $m > 0$ a parameter approximating the dimension of the data manifold in $\mathbb{R}^{dQ}$. The density estimator is formally given by $\rho(y) = \int \exp(-\|z - y\|^2/\tilde{\gamma}^2)\, d\nu(z)$, where $\tilde{\gamma} > 0$ is a bandwidth parameter (in general, different from $\gamma$ in (2)). The dimension parameter $m$ is determined numerically using the same procedure[40] (outlined below) as for tuning the bandwidth parameters $\gamma$ and $\tilde{\gamma}$. In the L63 experiments, we obtain a value $m \approx 2.06$ approximating the fractal dimension of the Lorenz attractor. Even though the L63 snapshot data $x_i \in \mathbb{R}^3$ contain full state information, we have used a long embedding window of $Q = 800$ samples at a $\Delta t = 0.01$ sampling interval. This has the effect of "biasing" the basis vectors $\phi_j$ towards approximate Koopman/transfer eigenvectors[33,37], thus improving the efficiency of the basis in approximating solutions to the generator eigenvalue problem.

In all generator calculations reported in this paper we tune the bandwidth parameters $\gamma$ and $\tilde{\gamma}$ automatically using a numerical procedure[40]. This involves computing the kernel sum $S(\gamma_l) := \sum_{i,j=Q-1}^{N-2} k_{\gamma_l}(\tilde{x}_i, \tilde{x}_j)$ on a logarithmic grid $\gamma_l$ of trial bandwidth parameters, and choosing $\gamma$ as the bandwidth parameter $\gamma_l$ that

maximizes the derivative $d\log S(\gamma_l)/d\log \gamma_l$ (estimated numerically by finite differences). See Algorithm 1 in Ref. [34] for pseudocode. The maximum value $\hat{m}$ of $d\log S(\gamma_l)/d\log S(\gamma_l)$ can be shown to be approximately equal to the dimension of the data manifold in $\mathbb{R}^{dQ}$ divided by 2. Based on that, in our generator calculations utilizing the bandwidth function in (6) we set the dimension parameter $m = \hat{m}/2$.

**Phase composites.** Here, we describe the procedure for constructing phase composites of the observables employed in Figs. 7 and 8. Let $Y : \Omega \to \mathbb{R}^{d'}$ be a target observable for compositing. For instance, in Fig. 7, $Y$ is either the global SST, zonal surface wind, or meridional surface wind anomaly field sampled at $d'$ gridpoints. Let $g : \Omega \to \mathbb{C}$ be a complex-valued index representing the phenomenon of interest for which composites are created. In Figs. 7 and 8, $g$ is equal to either the generator eigenfunctions $g_j$, or the lagged Niño 3.4 index $f_{\mathrm{nino}}$. We further let $\hat{\Omega} = \{\omega_{Q-1}, \ldots, \omega_{N-1}\}$ denote the set of states sampled along our dynamical trajectory (taking delay embedding with $Q$ delays into account), $S \in \mathbb{N}$ the number of phases, and $m$ an integer less than $\tilde{N}/S$ representing the number of samples in each phase (recall that $\tilde{N} = (N - Q + 1)/S$). We define $S$ "wedges" $W_1, \ldots W_S \subset \mathbb{C}$ in the complex plane by

$$W_j = \{z \in \mathbb{C} : |z| \geq a_j \text{ and } \arg z \in \Theta_j\}, \quad j = 1, \ldots, S,$$

where $\Theta_j = [2\pi(j-1)/S, 2\pi j/S)$, and $a_j$ is the $m$-th largest modulus of the complex numbers in the set $\{g(\omega) : \omega \in \hat{\Omega} \text{ and } \arg \omega \in \Theta_j\}$.

The sets $W_1, \ldots, W_S$ represent $S$ "phases" of an oscillatory process represented by $g$. In addition, we define a phase $W_0 = \mathbb{C} \setminus \bigcup_{j=1}^{S} W_j$ associated with the states in $\Omega$ for which the process represented by $g$ is considered inactive. For each phase $W_j$ we define the associated *phase composite* as the vector $\mathbb{Y}_j \in \mathbb{R}^d$ given by the average $\mathbb{Y}_j = \sum_{\omega_i \in W_j} Y(\omega_i)/|W_j|$, where $|W_j|$ denotes the number of elements of $W_j$. Note that $|W_1| = \ldots = |W_S| = m$ and $|W_0| = \tilde{N} - mS$.

We can interpret the phase composites $\mathbb{Y}_j$ as values of the conditional expectation of the observable $Y$ with respect to the partition $\{W_0, \ldots, W_S\}$ of $\mathbb{C}$ induced by the complex-valued index $g$. For that, note that the partition induces a discrete variable $\pi : \Omega \to \{0, 1, \ldots, S-1\}$, where $\pi(\omega) = j$ if and only if $\omega$ lies in $W_j$. We define $\mathbb{Y} : \Omega \to \mathbb{R}^{d'}$ as a discrete observable representing the empirical conditional expectation of $Y$ given $\pi$, i.e.,

$$\mathbb{Y} = \mathbb{E}_{\mu_N}(Y|\pi) = \sum_{j=0}^{S} \mathbb{Y}_j \chi_j.$$

Note that $\mathbb{Y}$ is a discrete observable satisfying $\mathbb{Y}(\omega) = \mathbb{Y}_j$ whenever the eigenfunction value $g(\omega)$ lies in $W_j$ for the state $\omega \in \Omega$.

**Mode reconstruction.** Our approach for computing spatiotemporal mode reconstructions (e.g., as shown in Fig. 10) is closely related to the reconstruction procedure in SSA[55], with appropriate modifications to take into account the facts that eigenfunctions of evolution operators may be (i) complex-valued; and (ii) non-orthogonal. Let $Y : \Omega \to \mathbb{R}^{d'}$ be a target observable for reconstruction as in the previous section. For instance, in Fig. 10, the target observable $Y$ is the Niño 3.4 region-averaged SST anomaly, which is a scalar with $d' = 1$, but $Y$ can also be vector-valued, for example when one reconstructs the original input data and sets $Y = X$.

Let $\langle \cdot, \cdot \rangle$ denote the inner product of $H$, $\langle f_1, f_2 \rangle = \int_\Omega \bar{f}_1 f_2\, d\mu$, and let $g_j'$ denote an element of the biorthonormal basis of the $\{g_i\}$, $\langle g_j', g_i \rangle = \delta_{ji}$. A procedure for constructing the biorthonormal set $\{\boldsymbol{g}', \ldots, \boldsymbol{g}_{L-1}'\}$ to $\{\boldsymbol{g}_0, \ldots, \boldsymbol{g}_{L-1}\}$, satisfying $\boldsymbol{g}_i'^\dagger \boldsymbol{g}_j = \delta_{ij}$ is to (i) form the $L \times L$ Gram matrix $\boldsymbol{G}$ with $G_{ij} = \boldsymbol{u}_i^\dagger \boldsymbol{u}_j$; (ii) compute $\boldsymbol{u}_i' = \boldsymbol{G}^{-1} \boldsymbol{u}_i$; and (iii) form the linear combination $\boldsymbol{g}_i' = \sum_{k=0}^{L-1} u_{ki}' \boldsymbol{\phi}_k$.

For each eigenfunction $g_j$ and lag $q \in \mathbb{Z}$, we define the complex-valued spatial pattern $A_j^{(q)} \in \mathbb{C}^{d'}$ given by projection of $P^{q\Delta t} Y = (U^{q\Delta t})^* Y$ onto generator eigenfunction $g_j$; formally,

$$A_j^{(q)} = \langle g_j', P^{q\Delta t} Y \rangle = \lim_{N \to \infty} \frac{1}{N} \sum_{i=Q-1}^{N-1} \overline{g_j'(\omega_i)} Y(\omega_{i-q}). \tag{7}$$

Numerically, we approximate $A_j^{(q)}$ by projecting the samples $y_0, \ldots, y_{N-1}$, $y_i = Y(\omega_i)$, of the target observable, lagged by $q$ steps, onto the dual eigenvector $\boldsymbol{g}'$, viz.

$$\hat{A}_j^{(q)} = \frac{1}{\tilde{N}} \sum_{i=Q-1}^{N-1} \overline{g_{ij}'} y_{i-q}.$$

It is worthwhile noting that for $q = 0$ the spatial patterns $\hat{A}_j^{(0)}$ are analogous to Koopman modes employed in data-driven Koopman operator techniques[12,13,17]. The patterns $\hat{A}_j^{(0)}$ can thus be thought of as time-shifted Koopman modes.

Next, we define an approximate projection of the target observable $Y$ onto the eigenfunction $g_j$, namely $\tilde{Y}_j : \Omega \to \mathbb{C}^{d'}$, by multiplication of $A_j^{(q)}$ with $U^{q\Delta t} g_j$,

followed by averaging over the delay-embedding window[55],

$$\tilde{Y}_j = \frac{1}{Q} \sum_{q=-Q/2}^{Q/2} A_j^{(q)} U^{q\Delta t} g_j.$$

Numerically, $\tilde{Y}_j$ is approximated by the spatiotemporal pattern $\hat{Y}_j = (\hat{y}_{j(Q-1)}, \dots, \hat{y}_{j(N-1)}) \in \mathbb{C}^{d' \times \tilde{N}}$, where

$$\hat{y}_{ji} = \frac{1}{Q} \sum_{q=-Q/2}^{Q/2} \hat{A}_j^{(q)} g_{i+q,j}, \qquad (8)$$

approximates $\tilde{Y}_j(\omega_i)$. Note that $\tilde{Y}_j$ can be equivalently expressed as

$$\tilde{Y}_j = \frac{1}{Q} \sum_{q=-Q/2}^{Q.2} \langle U^{q\Delta t} g_j', Y \rangle U^{q\Delta t} g_j,$$

from which we can interpret $\tilde{Y}_j$ as a projection of the observable $Y$ onto an order-$Q$ Krylov subspace generated by eigenfunction $g_j$. Adopting standard terminology from climate science, in the main text we refer to the reconstructed patterns $\hat{Y}_j$ as *modes*, though it should be kept in mind that these patterns are different from Koopman modes in that they have both spatial and temporal character.

The individual modes $\tilde{Y}_j$ can be combined into sum modes by choosing an index set $J = (j_1, \dots, j_l)$ and defining $\tilde{Y}_J = \sum_{k=1}^{l} \tilde{Y}_{j_k}$. Similarly, in the empirical setting, we define $\hat{Y}_J = \sum_{k=1}^{l} \hat{Y}_{j_k}$. Note that $\tilde{Y}_J$ (resp. $\hat{Y}_J$) is real whenever $J$ consists of indices of pairs of complex-conjugate eigenvalues $\lambda_j$ (resp. $\hat{\lambda}_j$). In Fig. 10, we show reconstructions using index sets $J$ representing various complex-conjugate pairs of ENSO and ENSO combination modes associated with generator eigenfunctions.

## Data availability

The CCSM4 data analyzed in this study are available at the Earth System Grid repository under accession code https://www.earthsystemgrid.org/dataset/ucar.cgd.ccsm4.joc.b40.1850.track1.1deg.006.html. The ERSSTv4 and NCEP reanalysis data are available at the National Centers for Environmental Information repositories, under accession codes https://www.ncdc.noaa.gov/data-access/marineocean-data/extended-reconstructed-sea-surface-temperature-ersst-v4 and https://psl.noaa.gov/data/gridded/data.ncep.reanalysis.html, respectively. The processed data are available from the corresponding author on reasonable request. The processed data can also be generated by running the MATLAB code in the repository https://doi.org/10.5281/zenodo.5508376 or https://doi.org/10.5281/zenodo.5511734.

## Code availability

MATLAB code implementing the numerical techniques and reproducing the generator results described in the paper is available at https://doi.org/10.5281/zenodo.5508376. See the file /pubs/FroylandEtAl21_NatComms/README in the code repository for additional information. Code for the transfer operator computations in the Lorenz and ERSSTv4 examples is available at https://doi.org/10.5281/zenodo.5511734.

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

## Acknowledgements

This research was initiated during a 2-week visit of D.G. to UNSW in 2018, supported by GF's Future Fellowship, and further developed during a 5-week visit by G.F. to NYU in 2019, supported by the UNSW Faculty of Science's and School of Mathematics and Statistics' Special Studies Program, and an ARC Discovery Project. G.F. also thanks the Courant Institute for hospitality during this visit. D.G. received support from NSF grants 1842538 and DMS 1854383 and ONR YIP grant N00014-16-1-2649. BRL and MP received support from NSF grant 1842543. J.S. received support from NSF grant 1842538. J.S. also acknowledges support from the core funding of the Helsinki Institute for Information Technology (HIIT) and the Institute for Basic Sciences (IBS), Republic of Korea, under IBS-R028-D1.

## Author contributions

All authors contributed to the study conception and design. Material preparation, data collection and analysis was performed by all authors. The first draft of the manuscript was written by G.F., D.G., and B.L., and all authors commented on previous versions of the manuscript. All authors read and approved the final manuscript.

## Competing interests

The authors declare that they have no competing interests.

## Additional information

**Peer review information** *Nature Communications* thanks Valerio Lucarini and the other anonymous reviewer(s) for their contribution to the peer review this work. Peer reviewer reports are available.

