## [Peer Review File · Nature Communications]

Reviewers' Comments:

Reviewer #1:

Remarks to the Author:

Dear authors,

I have read your manuscript with great interest and I definitely support the use of methods borrowed from dynamical systems theory to better understand the mechanisms underpinning the complexity of the climate system. The paper is informative, is written with good clarity, and reflects the skill and competence of the authors. I am puzzled, through by some issues below

1) The author seem to use methods that relevant for autonomous system is a clearly non-autonomous setting; indeed, they discover signature of the seasonal cycle (and of climate change in the case of observations). Is it formally correct to proceed as currently done in the paper? How would they would otherwise proceed to remove the seasonal cycle/trends?

2) Why using a delay embedding of 48 months and not 12 or 120 months? It is also worrying to see that 48 months coincides with the frequency of the ENSO-like mode they find.

3) Regarding the ENSO-like mode. The authors correctly criticise the use of indices etc as empirical and sometimes potentially misleading. On the other side, they find a mode and then they call is ENSO because, so to say,, looks like ENSO. Hence, subjectivity is still there, in some sense.

4) The real part of the eigenvalues seem to be very different in the model vs observations, why? Also, why reporting the real part in arbitrary units, whereas they should have the same units as the imaginary parts?

5) I understand that ENSO combination modes describe the Interannual variability of ENSO. What about - in observations - possible trends of the ENSO mode?

6) I understand the merit and elegance of rectifying the life cycle of ENSO. But what do we exactly learn from this? What is the advantage? One could argue, in fact, that the fact that the ENSO cycle is not symmetric is interesting, because the processes of onset of the two phases of ENSO are different.

All in all, I would like the authors to convince me more a) of the robustness of their procedure, and b) that what they achieve is not *only* a very elegant description of data, but maybe stress more what is predictive about it.

This recent publication might provide further support for the authors' methodology:

M. Santos-Gutierrez et al., Reduced-order models for coupled dynamical systems: Data-driven methods and the Koopman operator, Chaos in press (2021) doi: 10.1063/5.0039496 - arXiv:2012.01068

It shows the close connection between Mori-Zwanzig projection, slow/fast modes separation, eigenvalues/eigenvectors of Koopman operator, and data-driven methods in the form of multilevel markov models/empirical model reduction methods.

Maybe it would help to write the paper is a somewhat less descriptive and more critical way. I would also recommend the authors to be a bit more concise. But of course I am well aware these are somewhat generic suggestions.

Best Regards,

Valerio Lucarini

Reviewer 2

This paper presents a novel analysis of El Nino Southern Oscillation (ENSO) based on spectral operator-theoretic methods (Koopman and transfer operators). The main method relies on the computation of dominant eigenfunctions of the Koopman operator generator, using finite-dimensional approximations obtained with kernel-based techniques. As demonstrated with synthetic and observation datasets, the proposed approach offers superior performance over classical approaches (climatic indices), e.g. in terms of phase identification in the ENSO cycle (dynamical rectification, phase equivariance). The approach is also capable of extracting combination modes between ENSO and the annual cycle that can be used to capture specific events, as well as non stationary modes that could be interpreted as the effect of climate change.

Although based on existing theory and methods, the proposed approach applied to the analysis of ENSO is novel and of great interest. The manuscript is also well-organized and well-written. I recommend the paper for publication but have some comments that the authors should address.

Methods. Two methods are presented, which are respectively based on the transfer operator and the (generator of) the Koopman operator. However, it seems that the results presented in the paper are mostly related to the latter. This should be clarified. Moreover the relevance of the method based on the transfer operator should be discussed, with possibly a comparison between both methods and guidelines to decide which method to use in future applications.

Ergodicity. The authors should discuss the relevance of the ergodicity assumption, which is key in the proposed approach.

Perturbed eigenvalues. Since the eigenvalues of the perturbed operator P_ε can be perturbed both in modulus and argument, R_ε should be complex-valued (p.6 and p.9), with $|R_\varepsilon - 1| \approx 0$.

Climate change. The fact that some generator eigenvalues capture a trend related to climate change seems to be a bold statement that should be further discussed and supported by additional analysis. Also shouldn't we (naively?) expect eigenvalues with positive real part in this case?

Numerical error. It is supposed that the difference between computed eigenvalues and theoretical values might be due to numerical approximation errors (finite-difference). This

assumption should be verified on synthetic data characterized by a shorter sampling period.

Rectified cycles. The idea of rectifying cycles from eigenfunctions should be associated to the work by I. Mezic and his followers. The connection to phase reduction methods could also be mentioned.

Phase composites. The notion of phase composites should be clearly defined (even if trivial). The method to compute them should also be detailed.

Modes. The last section on mode reconstruction is difficult to follow and some explanations on computation should be given in the main text. In particular, it is not clear how the results of Fig. 10 are obtained (e.g. which index set J is chosen). Speaking of modes, one would expect a spatial (and not a temporal) representation. The authors may also wish to discuss the relevance of computing spatial modes in the context of ENSO analysis.

Other. Typo in p.7 l.7 (by at a rate); incorrect reference to panels in the caption of Fig. 5; typo in p.18 l.25 (of the of the).

Reviewer 3

This article applies an algorithm based on transfer and Koopman operator to extract information about periodic components of climate dynamics from data. This application is demonstrated to exhibit good accuracy.

What I'd like to see more is a cleaner description of the main contribution and innovation. Is it about a new scientific discovery or a new data analysis methodology? I'd like to ask about the former, because experiments conducted are mainly for demonstrating the effectiveness of the algorithm as it reproduces known scientific knowledge, but I had trouble finding new science. I'd like to ask about the latter, because I was unsure, based on the current manuscript, what is new about the algorithm. How does it differ from the ones in the existing literature? The algorithm uses a good number of heuristics which are not theoretically justified in this article, and readers are frequently referred to the literature for more details, and thus I hope that a clearer description of the algorithmic innovation could be provided.

Since the problem this article considered is both important and challenging, I think it suffices to have just one of these two possible contributions (scientific and algorithmic). As Nature Communications is a top journal, I think discretion should be exercised, and therefore recommend a major revision for now.

Here is a couple more comments/questions:

- Page 5: the requirement that 'the observables are slowly decaying' sounds contradictory to 'cyclicity'. Please elaborate.
- Page 22: Gaussian kernel is used but it is not scaling invariant. Is there an implicit assumption that the data is almost isotropic?
- I also would like to understand the parameter tuning. To apply the method to extract scientific knowledge from a new data set, it is likely that little knowledge is known about what to expect. In this case, how does the parameter tuning work?

Spectral Analysis of Climate Dynamics with Operator-Theoretic Approaches Review Response

Gary Froyland, Dimitrios Giannakis, Benjamin Lintner, Maxwell Pike, Joanna Slawinska

July 28, 2021

We thank the Reviewers and Editors for their careful consideration and constructive comments on the manuscript. The principal changes made in response to these comments are as follows:

- We have revised and expanded the section “ENSO diversity”.
- We have clarified aspects of the ergodicity/autonomicity assumptions of our methodology, construction and interpretation of phase composites, and reconstruction of spatiotemporal modes.
- We have performed sensitivity analyses of our results with respect to the length of the delay embedding window and finite-difference approximation of the generator.

Below is a point-by-point response to the Reviewers’ comments, using an italic font to indicate text copied verbatim from their reports. Changed or added text in the revised manuscript is highlighted in blue.

1 Response to Reviewer #1

1) The author seem to use methods that relevant for autonomous system is a clearly non-autonomous setting; indeed, they discover signature of the seasonal cycle (and of climate change in the case of observations). Is it formally correct to proceed as currently done in the paper? How would they would otherwise proceed to remove the seasonal cycle/trends?

The CCSM model dynamics $\Phi^t : \Omega \rightarrow \Omega$ is autonomous, so it is formally correct to apply the method to the CCSM data. In this case, the seasonal cycle is not a signature of non-autonomicity in Φ^t , and we therefore do not need to remove the seasonal cycle. It is true that the ERSST data has the climate warming trend and that strictly speaking our autonomous methodology does not formally apply here. Nevertheless, our spectral decomposition separates the trend (corresponding to a real eigenvalue) from the periodic cycles (corresponding to complex eigenvalues). The periodic cycles are by definition trendless. This decomposition effectively removes the trend in the same way that seasonal cycles are “removed” by appearing as separate cycles (with separate eigenvalues/eigenfunctions) to the ENSO cycle.

It should also be noted that stationarity/autonomous dynamics is a commonly made tacit assumption in many analyses of *observational* climate data where only one trajectory is available (as opposed to ensemble data from climate models providing access to multiple trajectories). For instance, a standard EOF analysis of the ERSST data would implicitly assume stationarity and ergodicity to ensure (i) well-definition of the covariance operator associated with the observation map $X : \Omega \rightarrow \mathbb{R}^d$, and (ii) consistency of its data-driven approximation. Analogously, in our approach, autonomicity and ergodicity gives us (i) well-defined groups of Koopman/transfer evolution operators, and (ii) consistency of data-driven approximations.

A fairly common approach in the literature is to “detrend” the data (typically by estimation and removal of a linear trend), but this does not eliminate the non-stationarity of the system, and carries a risk of introducing biases. We believe that an advantage of our approach is that it is capable of extracting trendless modes in ERSST without ad hoc detrending of the data. We have added text in the fourth and third paragraphs of pp. 8 and 13, respectively, addressing these points.

2) *Why using a delay embedding of 48 months and not 12 or 120 months? It is also worrying to see that 48 months coincides with the frequency of the ENSO-like mode they find.*

For state estimation, a single lag of 12 months is sufficient, as used in the transfer operator computations. The 48 months used in the generator computations is not for state estimation, but rather for numerical purposes to bias the ansatz space for generator eigenfunctions toward containing eigenfunctions of the true generator.

In order to verify the robustness of our numerical ENSO frequencies, we have performed generator computations with monthly lags and delay embedding windows in the range 1–16 years. The resulting ENSO eigenfrequencies and eigenperiods are listed in the table below.

Embedding window Q (months)	ENSO eigenfrequency (cycles/year)	ENSO eigenperiod (years)
12	0.2512	3.9810
24	0.2502	3.9968
48	0.2507	3.9895
96	0.2535	3.9451
192	0.2543	3.9324

The eigenfrequency results are consistent to within two significant digits across the examined range of embedding windows. We should note that the corresponding eigenfunctions exhibit more sensitivity to the choice of embedding window, becoming increasingly concentrated in the frequency domain as the embedding window increases (as expected theoretically). See Fig. 9 in Slawinska & Giannakis (2017), *J. Climate*, doi:10.1175/JCLI-D-16-0176.s1 for a discussion of this behavior in a related context. We have added text discussing aspects of robustness, including the aforementioned numerical experiments, in the first and third paragraphs of p. 10.

3) *Regarding the ENSO-like mode. The authors correctly criticise the use of indices etc as empirical and sometimes potentially misleading. On the other side, they find a mode and then they call it ENSO because, so to say,, looks like ENSO. Hence, subjectivity is still there, in some sense.*

First, we should note that ENSO does not have a universally-accepted, quantitative definition, and is always defined with respect to an index. In the context of our spectral analysis approach, the subjectivity, if any, in associating particular transfer/generator eigenfunctions with ENSO is limited. We compute the transfer operator or generator spectrum and select the complex eigenvalue with largest real part (i.e., smallest decay rate) and frequency closest to the approximate ENSO frequency. We refer to this eigenfunction as the fundamental ENSO eigenfunction. This is an unambiguous choice as there is only a single option; see Fig. 3 and the third paragraph on p. 9 in the manuscript.

We apply our analysis to the fundamental ENSO eigenfunction and we report what comes out, which turns out to represent the ENSO cycle well on a number of measures. These measures include the SST and wind fields in Figs. 7 and 8. Phase composites of other climatic variables, such as sea surface height, surface air temperature, and precipitation are also found to be consistent with the ENSO cycle. While we were not able to include these results in the manuscript due to space limitations, we include representative composites of SSH, SAT, and precipitation (in addition to the SST and surface wind composites included in the manuscript) in Figure 1 below. There, one can observe for instance that the SSH and precipitation composites exhibit broadly consistent features with the corresponding SST and surface wind patterns throughout the ENSO cycle.

4) *The real part of the eigenvalues seem to be very different in the model vs observations, why? Also, why reporting the real part in arbitrary units, whereas they should have the same units as the imaginary parts?*

We have rescaled the axes to make the two spectra more easily comparable. The observational data are sparser and noisier, so we expect larger decay rates compared to the model data. This is borne out in Fig. 3, where the real parts of the generator eigenvalues are more negative for the ERSST data. We have added a remark about these points in the second paragraph of p. 10.

The reason for labeling the units of the real parts of the eigenvalues as “arbitrary” was to minimize risk that readers interpret the values of the corresponding decay rates as having physical significance in the

Figure 1: Phase composites of sea surface temperature (SST; colors), sea surface height (SSH; colors), surface air temperature (SAT; colors), and surface wind anomalies (arrows) associated with the generator eigenfunction extracted from CCSM4.

context of climate dynamics. We realize however that our usage of the label “arbitrary units” could have created confusion by suggesting that the axes for $\text{Re } \lambda_j$ and $\text{Im } \lambda_j$ are independently scaled (which is not the case), so we have removed it.

5) *I understand that ENSO combination modes describe the Interannual variability of ENSO. What about - in observations - possible trends of the ENSO mode?*

ENSO combination modes describe the *seasonal* variability of ENSO. The question that the reviewer raises about possible trend-ENSO combination modes is intriguing. In previous work [Giannakis & Slawinska (2018), *J. Climate*, doi:10.1175/JCLI-D-17-0031.1] we had identified (using a kernel based approach) decadal modes in CCSM4 with statistically significant correlation with the ENSO amplitude consistent with a decadal-ENSO combination modes. That analysis, however, employed long model integrations and a decadal delay embedding window to robustly extract decadal modes. The present analysis interval only spans approximately 12 ENSO cycles, and we believe that attempting to identify trend-ENSO combination modes with the currently available data and without further analysis (lying outside the scope of this project) would be speculative. As just mentioned, we believe this would be an interesting direction for future work.

6) *I understand the merit and elegance of rectifying the life cycle of ENSO. But what do we exactly learn from this? What is the advantage? One could argue, in fact, that the fact that the ENSO cycle is not symmetric is interesting, because the processes of onset of the two phases of ENSO are different.*

We have extracted a canonical ENSO cycle and we may represent that cycle according to real time or rectified time. In rectified time, it is clear that the ENSO cycle is asymmetric because La Niña appears earlier (in phase/angle space) around the cycle (Fig. 5). From rectification we also learn an estimate of the speed of the cycle in real time. If one did not have the rectified representation, it would be difficult to assign a local speed around the cycle. This notion of local speed may be useful for constructing reduced models. Finally, we suggest that rectification is an important conceptual construction that the climate community may find useful. We have added text related to these points in p. 2 and p. 15.

*All in all, I would like the authors to convince me more a) of the robustness of their procedure, and b) that what they achieve is not *only* a very elegant description of data, but maybe stress more what is predictive about it.*

Regarding robustness, we used two different data sets (CCSM and ERSST), and we used two different methods (transfer operator and generator) and we obtained very similar results. We have added text to explain this in the third paragraph of p. 10. Regarding prediction, we have characterized ENSO phases in a way that is more predictable, as evidenced in the equivariance plots in Fig. 9. In the process of extracting the slowest decaying cycle, we group together different historical time points in the “right” way, leading to greater predictability. It is possible that the reason for some of the difficulties with ENSO prediction is that existing characterizations of ENSO through scalar indices such as Niño3.4 partially mix together unrelated ENSO phases, leading to rapid divergence of these “false” groupings. We have added a remark on these points in the first paragraph of p. 21.

This recent publication might provide further support for the authors’ methodology: M. Santos-Gutierrez et al., Reduced-order models for coupled dynamical systems: Data-driven methods and the Koopman operator, Chaos in press (2021) doi: 10.1063/5.0039496 - arXiv:2012.01068. It shows the close connection between Mori-Zwanzig projection, slow/fast modes separation, eigenvalues/eigenvectors of Koopman operator, and data-driven methods in the form of multilevel Markov models/empirical model reduction methods.

Thank you for pointing out this reference. It is somewhat different in focus to the objectives of our paper, which is more about diagnostics and feature extraction than reduced order modeling. In this paper, our constructions are concerned with extracting cyclic behavior rather than producing a complete reduced predictive model. In particular, we do not assume timescale separation.

There is a considerable literature on reduced-order modeling, including operator-theoretic approaches, which is tangentially related to our work. However, on balance we felt that we had to draw the line somewhere, and after careful consideration we decided not to include this group of literature.

2 Response to Reviewer #2

Methods. Two methods are presented, which are respectively based on the transfer operator and the (generator of) the Koopman operator. However, it seems that the results presented in the paper are mostly related to the latter. This should be clarified. Moreover the relevance of the method based on the transfer operator should be discussed, with possibly a comparison between both methods and guidelines to decide which method to use in future applications.

We agree that additional clarity on which method to use would be warranted. The main difference between the two approaches is discrete versus continuous time. Since we work with unitary dynamics on L^2 , one can switch between discrete-time Koopman and transfer operators by taking operator adjoints, $U^* = P$. Similarly, in continuous time, one can switch between the generators of the Koopman and transfer evolution groups by multiplication by -1 .

We separated the discrete versus continuous time methods for clarity and to present the reader with different options for numerical approximation. Most results were presented for the generator to avoid unnecessary duplication of images and reporting of results. Both methods gave very similar results in terms of extracting ENSO modes on both datasets. We have added text in the fifth paragraph of p. 6 making the distinction between discrete- and continuous-time approaches and the duality relationships between Koopman and transfer operators explicit.

Ergodicity. The authors should discuss the relevance of the ergodicity assumption, which is key in the proposed approach.

Please see our response to comment 1 of Reviewer 1 (and corresponding changes in the manuscript), who asks a closely related question about autonomous vs. non-autonomous dynamics. For autonomous dynamics, ergodicity is a common and mild additional assumption. This assumption is tacitly made by virtually every data-driven technique we are aware of that operates on data sampled on a *single* dynamical trajectory (as in the case of observational climate data).

Perturbed eigenvalues. Since the eigenvalues of the perturbed operator P_ϵ can be perturbed both in modulus and argument, R_ϵ should be complex-valued (p.6 and p.9), with $|R_\epsilon - 1| \approx 0$.

We have clarified in the text (p. 7) that the phase is absorbed into the argument to retain a real modulus R_ϵ . Our notation now reads $\Lambda_{\pm 1}^{(\epsilon)} = R_\epsilon e^{\pm i\alpha_\epsilon} \approx R_\epsilon e^{\pm i\alpha}$. We have made a similar correction in the continuous-time case in p. 7, $\lambda_{\pm 1}^{(\epsilon)} = -r_\epsilon \pm i\alpha_\epsilon \approx -r_\epsilon \pm i\alpha$, to ensure that the decay rate exponent r_ϵ is non-negative.

Climate change. The fact that some generator eigenvalues capture a trend related to climate change seems to be a bold statement that should be further discussed and supported by additional analysis. Also shouldn't we (naively?) expect eigenvalues with positive real part in this case?

Since our advection and diffusion operator V_ϵ is mass conserving, the generator cannot have eigenvalues with positive real part. Increasing values of observations in phase space is not related to the presence of unstable eigenvalues. We do not have a thorough theoretical explanation for the trend eigenvector at present; this will be a matter for future work. Nevertheless, the time series associated with the trend eigenfunction g_5 in Supplementary Figure 1 exhibits a number of accepted qualitative features of climate change in recent decades, including: (1) a rapid increase during the mid to late 1970s, early to mid 2000s, and early to mid 2010s, (2) a more gradual increase from 1980 to 2000, and (3) a warming “hiatus” during the mid 2000s to mid 2010s. These features are pointed out in the caption of the figure, as well as in p. 10 of the main text. As more quantitative metrics, we have computed correlation coefficients between g_8 and area-averaged anomalies of global surface air temperature (SAT) and Indo-Pacific SST. The correlation coefficients are found to be 0.78 and 0.83, respectively. We have included this information in the second paragraph of p. 10.

Numerical error. It is supposed that the difference between computed eigenvalues and theoretical values might be due to numerical approximation errors (finite-difference). This assumption should be verified on synthetic data characterized by a shorter sampling period.

Figure 2: Relative error, $|\nu_j^{\text{FD}} - \nu_j|/|\nu_j|$, of the leading four nonzero numerical eigenfrequencies, ν_j^{FD} , of the generator of a circle rotation (simple harmonic oscillation) as a function of the order of the finite-difference approximation scheme. The rotation period is equal to $T = 1$ (i.e., the angular frequency is $\alpha = 2\pi/T = 2\pi$), so the theoretical eigenfrequencies are $\nu_j = 2\pi j/\alpha = j$. The generator was approximated in a basis of kernel eigenfunctions with addition of diffusion as described in the manuscript. The sampling interval was $\Delta t = 1/12$ (i.e., analogous to monthly sampling of the seasonal cycle), and a total of $50 \times 12 = 600$ samples was employed (analogous to the ≈ 50 -year timespan of the ERSSTv4 analysis dataset). The finite-difference approximation type was central.

Since in analyses of climate data one does not typically have control of the sampling interval, we have examined the dependence of the numerical generator eigenvalues on the order of the finite difference approximation. Figure 2 below shows the relative numerical error in the leading four nonzero eigenfrequencies of the generator for a circle rotation (harmonic oscillator) of period 1, sampled at a sampling interval $\Delta t = 1/12$ (“1 month”) for 50 periods (“years”), as a function of the order of (central) finite-difference approximation. In these experiments, the numerical eigenfrequency ν_3 based on the fourth-order finite-difference approximation is analogous to the triannual frequencies in Fig. 3. The numerical results show that the numerical error can be reduced by increasing the finite difference order to 6 or 8. Since in this paper our focus is on lower frequencies than the triannual frequency (e.g., the interannual ENSO frequency $\nu_{\text{ENSO}} \approx 0.25$ cycles/year) we decided to keep the current results obtained with a fourth-order scheme.

The results from these experiments are outline in the first paragraph of p. 9, and are discussed in further detail in Methods (p. 28). We inform the reader that the error in the numerical triennial eigenfrequencies can be reduced by increasing the order of the finite-difference approximation of the generator. We also mention other contributing factors to errors in the eigenfrequencies, namely the Nyquist limit and the addition of diffusion (which will in general perturb the eigenvalues along both the real and imaginary axes).

Rectified cycles. The idea of rectifying cycles from eigenfunctions should be associated to the work by I. Mezic and his followers. The connection to phase reduction methods could also be mentioned.

In the beginning of p. 13, we have added a reference to further clarify that the arguments of exact Koopman eigenfunctions precess at a constant rate under the action of the Koopman operator, and references related to the use of the Koopman operator for other types of rectification.

Phase composites. The notion of phase composites should be clearly defined (even if trivial). The method to compute them should also be detailed.

We have added a subsection in Methods (pp. 29–30) describing the construction of phase composites. In the main text (last paragraph in p. 15) we indicate that the phase composites correspond to the value of the

conditional expectation of the observable of interest with respect to a discrete variable indexing the “wedges” in the complex plane associated with a particular eigenfunction/index.

Modes. The last section on mode reconstruction is difficult to follow and some explanations on computation should be given in the main text. In particular, it is not clear how the results of Fig. 10 are obtained (e.g. which index set J is chosen). Speaking of modes, one would expect a spatial (and not a temporal) representation. The authors may also wish to discuss the relevance of computing spatial modes in the context of ENSO analysis.

We have added text in the second paragraph of the subsection on ENSO diversity (p. 21–22) outlining how the reconstructions are produced. In addition, we now explicitly state the index sets J employed in the caption of Fig. 10. The approach that we use for reconstruction has been introduced and employed extensively in the context of SSA and EEOF analysis (see, e.g., Ref. 56 in the main text). In Methods, we have tried to describe the procedure from the perspective of operator-theoretic techniques.

Regarding our use of the term “modes”, we follow the general convention in climate science, where “mode” may be used to indicate any of a spatial, temporal, or spatiotemporal process. Please note that this convention is somewhat different from Dynamic Mode Decomposition (DMD), where the term “Koopman modes” is used specifically to refer to spatial patterns obtained by projection of the observed data onto Koopman eigenfunctions. We have added comments in pp. 21 (main text) and 30, 31 (Methods) to alert the reader of these facts. In climate science applications, the prevailing terminology for spatial patterns analogous to Koopman modes from DMD is Empirical Orthogonal Functions (EOFs), or Extended EOFs (EEOFs) in the context of singular spectrum analysis.

It should be noted that for each lag q , the spatial patterns $A_j^{(q)}$ in equation (7) (p. 30) can be thought of as time-shifted Koopman modes. Moreover, the procedure to obtain the reconstructed observables \tilde{Y}_j can be interpreted as a projection of the target observable Y onto the order- Q Krylov subspace generated by the (approximate) Koopman/transfer operator eigenfunction g_j , where Q is the number of Takens delays. We have added text in p. 30 to make these connections explicit.

Other. Typo in p.7 l.7 (by at a rate); incorrect reference to panels in the caption of Fig. 5; typo in p.18 l.25 (of the of the).

Thank you for catching these—the typos have been corrected.

3 Response to Reviewer #3

What I'd like to see more is a cleaner description of the main contribution and innovation. Is it about a new scientific discovery or a new data analysis methodology? I'd like to ask about the former, because experiments conducted are mainly for demonstrating the effectiveness of the algorithm as it reproduces known scientific knowledge, but I had trouble finding new science. I'd like to ask about the latter, because I was unsure, based on the current manuscript, what is new about the algorithm. How does it differ from the ones in the existing literature? The algorithm uses a good number of heuristics which are not theoretically justified in this article, and readers are frequently referred to the literature for more details, and thus I hope that a clearer description of the algorithmic innovation could be provided.

Since the problem this article considered is both important and challenging, I think it suffices to have just one of these two possible contributions (scientific and algorithmic). As Nature Communications is a top journal, I think discretion should be exercised, and therefore recommend a major revision for now.

Scientific Discoveries:

1. While cyclical constructions of ENSO have been previously performed, as in, e.g., Fig. 2 of Timmerman et al. (2018) (cited as Ref. 4 in our bibliography), our analysis uses only SST rather than a combination of SST and subsurface data related to the depth or temperature of the thermocline. This suggests that there is an imprint of information from subsurface variables to sequences of SST snapshots (“SST with memory”).

Aside from the generator-based ENSO index, which is an original contribution, our lagged construction of the Niño 3.4 index to create a Niño 3.4 cycle in Figs. 5 and 6 has to our knowledge not appeared in the literature before. We believe this is an important way to visualize scalar indices to emphasize cyclicity or lack thereof. We have noted on p. 13 that this construction is new.

2. While it has been known since the 1930s that the arguments of complex eigenfunctions precess at a constant rate under the Koopman operator, the explicit conjugacy we develop to “rectify” the ENSO cycle is new, and crucial to determining a local “speed” at different phases of the cycle. Our rectification is also an important conceptual construct for climate science. We added text in the introduction on p. 2 and on p. 15.
3. Our results highlight the possibility that existing methods of characterizing ENSO (such as scalar indices) may conflate unrelated parts of the cycle. This could contribute to difficulties with ENSO prediction, as shown in Fig. 9a,b (top), where poorly chosen groupings mix together unrelated ENSO phases, leading to rapid divergence of these “false” groupings. Our results suggest that ENSO may have a more significant cyclic component than previously realized. Indeed, we have characterized ENSO phases in a way that is more predictable; these are the equivariance plots in Fig. 9a,b (bottom). In the process of extracting the slowest decaying cycle, we group together different historical time points in the “right” way, leading to greater predictability of our computed ENSO phase. Please see Supplementary Figure 2 for a quantitative comparison of the predictability of ENSO phases associated with the Niño 3.4 index and generator eigenfunction. We have added text in the Introduction on p. 2 and the section titled “Phase equivariance” on p. 21.
4. Our analysis of the Niño 3.4 index in the subsection “ENSO diversity” shows how a small family of coherent modes can reconstruct diverse patterns of ENSO behavior. A result stemming from this analysis that has not been previously reported in the literature is the role of ENSO combination modes in generating ENSO events with multiple peaks. In addition, our analysis has identified an interannual mode with a 3-year characteristic timescale that contributes significantly to strong El Niño events of Eastern Pacific type (see Fig. 10). We believe that this result provides a more granular description of the role of ENSO modes with quasi-biennial timescales, which in previous studies have been associated with *both* Central and Eastern Pacific events (e.g., Ref. 87). In this revision, we have significantly rewritten the subsection on ENSO diversity to better highlight these contributions. Figure 10 has also been updated to better illustrate the role of the combination modes in recovering the “double-peaked” 1986/87 El Niño.

New methodologies:

1. The projection onto a cycle using level sets of the arguments of complex eigenfunctions is new. In fast/slow systems, previously Froyland et al. 2014 described a projection along level sets of a real eigenfunction as a means to project along fast fibers onto a slow manifold; we added text at the end of the first paragraph on p 11. This type of projection for complex eigenfunctions onto cycles is completely new (for example Fig. 2f and the section “Factoring out approximate cycles from eigenfunctions”).
2. When constructing the wedges in Fig. 6a,b,d,e, we restrict the wedges to include only data points for which the ENSO eigenfunction has large magnitude. While this emphasis on large-magnitude values has appeared in the literature in the context of coherent sets (e.g. Froyland et al. 2010, 2019), this crucial aspect has not been used for complex eigenfunctions and cycles before. We have added text to the last paragraph of p. 15.
3. Related to the item directly above, the testing of equivariance in Fig. 9 is a new and generally applicable way of visualizing the quality of the extracted cycle with respect to coherence.
4. The technique for approximating the transfer/Koopman operator with respect to the underlying invariant measure in the Methods section “Approximation of transfer and Koopman operators” is a new, simple, and effective approach for very high-dimensional systems. Moreover, it produces a nonnegative matrix representation, which is a further desirable property, not obtained by general Galerkin approaches. We added text in the Methods section on p. 25.

Page 5: the requirement that ‘the observables are slowly decaying’ sounds contradictory to ‘cyclicity’. Please elaborate.

For us, “slowly decaying” and “slowly decorrelating” are synonymous terms. We have added text to make this explicit in pp. 2 and 4. For the transfer operator (discrete time), the rate of decay is given by the magnitude of the eigenvalue and the cycle frequency is given by the argument of the eigenvalue. For the generator (continuous time), the rate of decay is given by the real part of the eigenvalue and the cycle frequency is given by the imaginary part of the eigenvalue. Thus, to find an eigenfunction corresponding to a slowly decaying observable we ask for the magnitude of a complex eigenvalue to be close to 1 (discrete time) or the real part of a complex eigenvalue to be close to 0 (continuous time). One can picture a slowly decaying cycle in phase space as a weakly stable focus; trajectories cycle around many, many times with the same frequency, very slowly converging to the equilibrium at the focus.

Page 22: Gaussian kernel is used but it is not scaling invariant. Is there an implicit assumption that the data is almost isotropic?

There is no assumption the data is almost isotropic. In the transfer operator calculations, the bandwidth of the Gaussian kernel in equation (1) is chosen large-enough so that nearest neighbor kernels overlap, which given our relatively frequent observations (monthly), means only a small constant bandwidth is needed. In the generator calculations, the variable-bandwidth kernel in equation (2) is, in fact, scale invariant by our choice of bandwidth function σ (see equation (6) and the displayed equation above it). That is, if the data is scaled by a constant factor c , i.e., $\tilde{x} \mapsto c\tilde{x}$, the bandwidth functions σ are chosen such that the ratio $\|\tilde{x}_i - y\|^2/\sigma^2(\tilde{x}_i, y)$ appearing in the argument of the exponential function in equation (2) remains unchanged. In fact, with our choices of bandwidth function, the kernels in (2) exhibit a stronger notion of scale invariance, namely asymptotic *local* scale invariance as $\gamma \rightarrow 0$ (which is equivalent to conformal invariance). Given a sufficiently smooth, invertible scaling function $c(\tilde{x})$, the transformation $\tilde{x} \mapsto c(\tilde{x})\tilde{x}$ imparts no changes to the eigenfunctions ϕ_j recovered from the kernel k_γ in equation (2) as $\gamma \rightarrow 0$; see Refs. 34 and 93 in our bibliography for more details.

I also would like to understand the parameter tuning. To apply the method to extract scientific knowledge from a new data set, it is likely that little knowledge is known about what to expect. In this case, how does the parameter tuning work?

The question that the Reviewer poses applies very broadly in computational science. As an analogy, suppose that one is interested in applying a fluid dynamical model to simulate fluid flow in a previously unexplored parameter regime (Reynolds number, Rayleigh number, etc.). Typically, there is theory ensuring that the numerical model converges in an appropriate asymptotic limit (e.g., under mesh refinement or under increasing spectral resolution), but for any given simulation one does not a priori know the errors incurred. Instead, the modeler relies on a combination of prior scientific knowledge about the phenomenon of interest and sensitivity analyses under variation of the model parameters.

More generally, the computational techniques employed in this work behave similarly to the above fluid dynamics analogy, with the difference that (i) we have to contend with statistical sampling errors (in addition to errors due to finite-rank approximation of operators encountered in a traditional numerical scheme for PDEs); and (ii) our methods involve meta-parameters such as the delay embedding window Q and bandwidth parameter ϵ (not too different from, e.g., numerical diffusion added to stabilize fluid dynamical models). As with many computational methodologies, our approach to tuning these parameter and extracting scientific knowledge is based on a combination of (i) prior physical knowledge (e.g., that ENSO has a $\simeq 4$ -year characteristic timescale); (ii) sensitivity analysis (see, e.g., the table in our response to comment 2 of Reviewer 1); and (iii) posterior analysis and physical interpretation of the results. Oftentimes, the steps (i)–(iii) would be applied iteratively, particularly in applications where there is limited prior knowledge.

Specifically for ENSO, in the discrete-time transfer operator approach, the choice of lag $\ell = 12$ months used the fact that we expected an approximately 4-year cycle for ENSO and chose the lag to be 1/4 of this cycle period, which is a classical choice in nonlinear time series analysis. For a cycle, usually an embedding dimension of 2 is sufficient and so we chose $Q = 2$ delays. An initial bandwidth parameter ϵ was chosen according to the mean nearest-neighbor distance (see the final paragraph of the “Approximation of transfer and Koopman operators” section in Methods). Because of the high-dimensional data and the

frequent data sampling of one month, this nearest neighbor is typically the next temporal data point, which is useful specifically for extracting cycles. The parameter ϵ may be further decreased as long as the eigenvalue 1 remains unit multiplicity and the numerical eigensolver converges. In the Lorenz-63 example, no prior knowledge of the system was used. No embedding was required (no use of ℓ or Q) and we simply slightly decreased ϵ from heuristic value discussed in Methods, stopping short of the point where ARPACK's eigenroutines no longer converged.

In the generator approach, the bandwidth parameter γ is tuned automatically; see the subsection "Bandwidth function and parameter tuning" in Methods and Ref. 34. Please see also the table in our response to comment 2 of Reviewer 1 and Figure 2 above for sensitivity analysis results for the delay embedding window Q and order of finite-difference approximation of the generator.

Reviewers' Comments:

Reviewer #1:

Remarks to the Author:

Dear authors,

thanks for your detailed reply to my comments and for the improved manuscript, which I find publishable in the current form. Let me, nonetheless, go through your replies because I would like to make some little further comments. I am following the enumeration you have used in your rebuttal letter.

1) The The CCSM model dynamics, even in a control run, is **not** autonomous, because, as you write correctly in the text, there is a perfectly periodic forcing due to the seasonal cycle. Indeed, the dynamics cannot be written as $\dot{x}=F(x)$, with x - clearly - very highly dimensional, but rather as $\dot{x}=F(x)+\epsilon G(x)\cos(\omega t)$. The system knows all too well when it is January and when it is July. Please clarify whether the methodology you propose can be applied for a periodically forced system.

I agree entirely with the pragmatic approach and viewpoint you describe in your reply to my comment. but I am suggesting to make clear to the reader what is rigorous and what pragmatically makes sense to do.

2) Thanks a lot for your clear explanation.

3) I am just saying that you provide a separate (and, I agree, very smart) definition of ENSO. With less sophistication, some scientists used specific EOFs/PCAs to define e.g. NAO rather than using the classic index.

4) Thanks a lot for your clear explanation.

5) Thanks a lot for your clear explanation.

6) Thanks a lot for your clear explanation. I concur with your comment.

Last point: It was just a suggestion for a reading of possible utility for interpreting the results.

Reviewer #2:

Remarks to the Author:

The authors have properly addressed all of my comments and have modified their manuscript accordingly. I have no further remark.

I recommend that this paper be accepted for publication.

Reviewer #3:

Remarks to the Author:

The authors have addressed my concerns.

Spectral Analysis of Climate Dynamics with Operator-Theoretic Approaches Review Response

Gary Froyland, Dimitrios Giannakis, Benjamin Lintner, Maxwell Pike, Joanna Slawinska

September 18, 2021

We thank the Reviewers and Editors for their careful consideration and constructive comments on the manuscript. The principal changes made in response to these comments are as follows:

- We have added text in the supporting online material, explaining how our methods are applicable to certain types of non-autonomous systems (including periodically forced systems) using state space augmentation.
- We have addressed the editorial requests in the attached document.

Below is a point-by-point response to the comments of Reviewer #1 comment, using an italic font to indicate text copied verbatim from their report. Changed or added text in the revised manuscript is highlighted in blue.

1 Response to Reviewer #1

*1) The CCSM model dynamics, even in a control run, is *not* autonomous, because, as you write correctly in the text, there is a perfectly periodic forcing due to the seasonal cycle. Indeed, the dynamics cannot be written as $\dot{x} = F(x)$, with x - clearly - very highly dimensional, but rather as $\dot{x} = F(x) + \epsilon G(x) \cos(\omega t)$. The system knows all too well when it is January and when it is July. Please clarify whether the methodology you propose can be applied for a periodically forced system.*

I agree entirely with the pragmatic approach and viewpoint you describe in your reply to my comment. but I am suggesting to make clear to the reader what is rigorous and what pragmatically makes sense to do.

The state space variables in the CCSM model include time (the variable t in the comment above); this is how the CCSM models knows to appropriately change its output with the seasons. Thus, in the notation of the above comment, we may write the CCSM state space variable as $\tilde{x} = (x, t)$, which are augmented with time. Now, indeed, we may represent CCSM dynamics as the autonomous system

$$\dot{\tilde{x}} = H(\tilde{x}),$$

which if we match the notation in the comment, is:

$$\dot{\tilde{x}} = (\dot{x}, \dot{t}) = (F(x) + \epsilon G(x) \cos(\omega t), 1) = H(\tilde{x}).$$

In our analysis, we do not observe \tilde{x} directly, but instead we observe SST images, which are functions of \tilde{x} . In the manuscript, these functions were denoted $X : \Omega \rightarrow \mathbb{R}^d$ and the full state was denoted ω (corresponding to the \tilde{x} in the current discussion). These SST images clearly contain the seasonal signature; that is, they are functions of \tilde{x} and not only of x . We have made this explanation in Supplementary Note 1.